# A double-stranded RNA binding protein enhances drought resistance via protein phase separation in rice

Huaijun Wang[1,3], Tiantian Ye[1,3], Zilong Guo ®[2], Yilong Yao[1], Haifu Tu[1], Pengfei Wang[1], Yu Zhang[1], Yao Wang[1], Xiaokai Li[1], Bingchen Li[1], Haiyan Xiong[1], Xuelei Lai[1] ✉ & Lizhong Xiong ®[1] ✉

Drought stress significantly impacts global rice production, highlighting the critical need to understand the genetic basis of drought resistance in rice. Here, through a genome-wide association study, we reveal that natural variations in *DROUGHT RESISTANCE GENE 9* (*DRG9*), encoding a double-stranded RNA (dsRNA) binding protein, contribute to drought resistance. Under drought stress, DRG9 condenses into stress granules (SGs) through liquid-liquid phase separation via a crucial α-helix. DRG9 recruits the mRNAs of *OsNCED4*, a key gene for the biosynthesis of abscisic acid, into SGs and protects them from degradation. In drought-resistant *DRG9* allele, natural variations in the coding region, causing an amino acid substitution (G267F) within the zinc finger domain, increase DRG9's binding ability to *OsNCED4* mRNA and enhance drought resistance. Introgression of the drought-resistant *DRG9* allele into the elite rice Huanghuazhan significantly improves its drought resistance. Thus, our study underscores the role of a dsRNA-binding protein in drought resistance and its promising value in breeding drought-resistant rice.

Drought stress induces significant yield loss in agricultural production, posing a formidable threat to global food security[1]. Rice, serving as a staple for over half of the world's population, faces severe limitations in cultivation and production due to drought[2]. Therefore, unraveling the genetic components contributing to rice drought resistance becomes a vital pursuit, so that the trait can be genetically improved. To understand the genetic basis of rice drought resistance, quantitative trait locus (QTL) mapping using recombinant inbred line (RIL) populations has been conducted. However, due to the complexity of the drought resistance which is associated with a variety of physiological parameters and controlled by multiple genes[3], the majority of the reported QTLs controlling drought resistance in rice remain as genetic loci[4,5], only a few genes, such as *DEEPER ROOTING 1* (*DRO1*)[6], have been cloned through fine mapping. Genome-wide association studies (GWAS) is a potent approach for the fine mapping of QTLs underlying complex traits, relying on the linkage disequilibrium (LD)[7]. In recent years, GWAS has been successfully applied to clone several drought-related genes, such as *ZmVPP1*[8], *TaNAC071-A*[9], and *DRESH8*[10]. Nonetheless, very few of these loci or genes have been successively used for drought resistance breeding in crops.

Plants have evolved sophisticated gene regulatory networks for responding to drought stress and the abscisic acid (ABA) signaling pathway is central to stress responses in plants[11]. 9-cis-epoxycarotenoid dioxygenase (NCED) functions as a critical rate-limiting enzyme in ABA biosynthesis[12,13], exerting a pronounced role in plant abiotic stress resistance[14]. Previously, many studies have reported that the expression of *NCED* is mainly regulated at the transcriptional level and participates in the plant responses to abiotic stresses[15,16]. However, it is unclear whether the ABA pathway genes including *NCED*s are regulated in other means such as posttranscriptional regulation.

[1]National Key Laboratory of Crop Genetic Improvement, Hubei Hongshan Laboratory, Huazhong Agricultural University, Wuhan, China. [2]Haixia Institute of Science and Technology, Fujian Agriculture and Forestry University, Fuzhou 350002, China. [3]These authors contributed equally: Huaijun Wang, Tiantian Ye. ✉e-mail: xuelei_lai@mail.hzau.edu.cn; lizhongx@mail.hzau.edu.cn

Posttranscriptional regulation is crucial for the precise and timely adjustment of plant transcripts in adverse environments[17]. This process involves the formation of interconnected mRNA-ribonucleoprotein (mRNP) complexes, including stress granules (SGs) and processing bodies (PBs)[18]. SGs are highly dynamic cytoplasmic assemblies found in diverse organisms[19]. The formation of SGs has been observed in plants under various stress conditions[20,21]. Previous studies showed that RNA-binding protein 47 (Rbp47), poly(A)-binding proteins (PABs), and oligouridylate binding protein 1 (UBP1) were SG components in *Arabidopsis thaliana* (Arabidopsis)[18]. SGs are important sites where translationally inactive mRNAs are sorted for degradation or stored for protection[22,23]. PBs, normally microscopically visible cytoplasmic granules under normal conditions, become larger when plants are stressed[24,25]. In Arabidopsis, PB markers including DECAPPING 1 (DCP1), DECAPPING 2 (DCP2), and VARICOSE have been idnetified[26]. In general, mRNAs targeted to PBs undergo degradation[27,28].

Double-stranded RNA (dsRNA) binding proteins have been found across various organisms, including plants. They orchestrate key roles in cellular processes including RNA interference, innate immunity, and post-transcriptional gene regulation[29–31]. In Arabidopsis, dsRNA binding proteins HYPONASTIC LEAVES 1 and DICER-LIKE 1 have been reported to play important roles in pre-miRNA processing in nucleus[32,33]. Nevertheless, it is still unclear whether there are other types of dsRNA binding proteins in plants, and whether dsRNA binding proteins are directly involved in the posttranscriptional regulation as SG components.

In this work, we identify a drought resistance gene, *DRG9*, encodes a dsRNA-binding protein by GWAS. DRG9 display diffuse cytoplasmic localization under normal conditions, but condense into SGs under drought stress. A highly conserved α-helix embedded within the intrinsically disordered region (IDR) of DRG9 is essential for the condensation function. DRG9 selectively binds to a spectrum of stress-related genes, including *OsNCED4*, and recruits *OsNCED4* mRNAs into SGs to protect them from degradation under drought conditions. A natural variation in the coding region of *DRG9* confers drought resistance by increasing the binding ability of DRG9 to *OsNCED4* mRNA and enhancing the mRNA stability. Furthermore, we demonstrate that a drought-resistant *DRG9* allele has been selected in the rice germplasms from drought-prone regions and has promising values in breeding drought resistant rice.

## Results

### Natural variations in DRG9 are associated with drought resistance

We previously identified significant association signals on chromosome 9 with GPAR_R (defined as green-projected area ratio under drought compared to control, one of the best image traits indicating drought resistance[34]) through GWAS. We performed a local linkage disequilibrium (LD) analysis to identify candidate genes around the lead single nucleotide polymorphism (SNP) and discovered 24 non-transposon coding genes located in candidate region (Supplementary Fig. 1a). By comparing the expression levels of the 24 genes under normal condition and drought stress, we found that *LOC_Os09g25430* (*ORF23*) exhibited strongly induced expression by drought stress (Supplementary Fig. 1b and Supplementary Data 1). In addition to *ORF23*, we also selected *ORF4*, *ORF6*, *ORF16*, *ORF19* and *ORF22* as candidate genes based on GWAS results and gene annotation information in our preliminary analyses. However, their mutants showed no significant drought resistance phenotype compared with the wild-type control (Supplementary Fig. 1c, d), thus *ORF23* was chosen as the candidate gene (designated as *DRG9* hereafter). Protein sequence alignments of DRG9 and its homologs revealed that DRG9 is conserved in monocot crops (Supplementary Fig. 1e). Gene-wise association analysis further validated the association signals including a single

nucleotide polymorphism (SNP) in the promoter (SNP2981, $P < 10^{-4}$) and three SNPs (SNP0823, SNP0824 and SNP1066, $P < 10^{-3}$) located in the third exon of *DRG9*. The three SNPs in the exon showed strong linkage disequilibrium (LD; $R^2 > 0.8$) with SNP2981 (Fig. 1a). Based on the genotypes of the four significant SNPs, the 503 rice accessions were primarily classified into two haplotypes (Fig. 1b). Notably, SNP1066 changed the 186th amino acid from asparagine (Asn) to serine (Ser), while SNP0823 and SNP0824 resulted in a substitution of the 267th phenylalanine (Phe) by glycine (Gly) (Fig. 1c). Germplasms carrying haplotype 1 (Hap1) exhibited significantly higher GPAR_R values compared to those carrying Hap2 ($P = 2.54 \times 10^{-4}$) (Fig. 1d). As a result, we designated Hap1 as the drought-resistant (DR) haplotype and Hap2 as the drought-sensitive (DS) haplotype for subsequent analyses. The expression pattern analysis by RT-qPCR showed that *DRG9* was expressed in various tissues and organs, but exhibited predominantly high levels in leaves at the tillering stage (Supplementary Fig. 1f). The expression of *DRG9* was induced by drought stress, and the levels increased along with the increase of drought severity (Supplementary Fig. 1g). To assess whether SNP2981 in the promoter caused any divergence in *DRG9* expression level, we investigated transcript abundance in 71 *DRG9*-DR accessions and 222 *DRG9*-DS accessions under normal and drought conditions (Supplementary Data 2). We found no significant difference in *DRG9* expression levels between the two haplotypes (Fig. 1e), suggesting that the difference in drought resistance between the two haplotypes is likely not caused by variation at transcriptional level.

### DRG9 positively regulates drought resistance in rice

To verify the functional involvement of *DRG9* in rice drought resistance, we generated *drg9* mutants through CRISPR/Cas9-mediated gene editing, and three independent frame-shift mutants (*drg9−26*, *drg9−29* and *drg9−33*) with 1 bp insertion in the second exon were obtained (Supplementary Fig. 2a). The *drg9* mutants was more sensitive to drought stress compared to the wild-type ZH11 at the seedling stage (Fig. 2a, b). Subsequently, we generated transgenic rice lines overexpressing the *DRG9* coding sequence from Nipponbare (*DRG9^Nip*) in KY131 (a widely planted rice variety in Northeast China) and three independent lines with high *DRG9* expression level were used for drought resistance assay (Supplementary Fig. 2b). These *DRG9* over-expression (OE) lines exhibited enhanced resistance to drought stress at the seedling stage compared to the wild-type KY131 (Fig. 2c, d). We further assessed drought resistance of the *drg9* mutants and OE lines at the reproductive stage. Under drought stress, while most leaves of the wild-type ZH11 remained green and alive, the leaves of the mutants were dried or almost dead, with markedly lower GPAR values than ZH11 (Fig. 2e, f). After recovery, the *drg9* mutants displayed a significant reduction in seed-setting rate and grain yield per plant compared to ZH11 (Fig. 2g, h). During the drought stress for the *DRG9* OE lines, while the leaves of wild-type KY131 were dried or almost dead, the *DRG9* OE lines remained green and alive and exhibited significantly higher GPAR values (Fig. 2i, j). After recovery, the *DRG9* OE lines exhibited a significant increase in seed-setting rate and grain yield per plant compared to KY131 (Fig. 2k, l). Under normal conditions, no significant difference was observed in seed-setting rate or grain yield per plant between the *drg9* mutants (or *DRG9* OE lines) and their respective wild-type plants (Supplementary Fig. 2c–f). Collectively, these results suggested that *DRG9* plays a positive role in drought resistance.

### DRG9 condensed into dynamic cytoplasmic granules under drought stress

To determine the subcellular localization of DRG9, we generated transgenic plants expressing DRG9^Nip-YFP fusion protein under the control of the maize ubiquitin promoter in the *drg9* mutant background, and found that the drought-sensitive phenotypes of the *drg9* mutant were rescued by the expression of DRG9^Nip-YFP

 

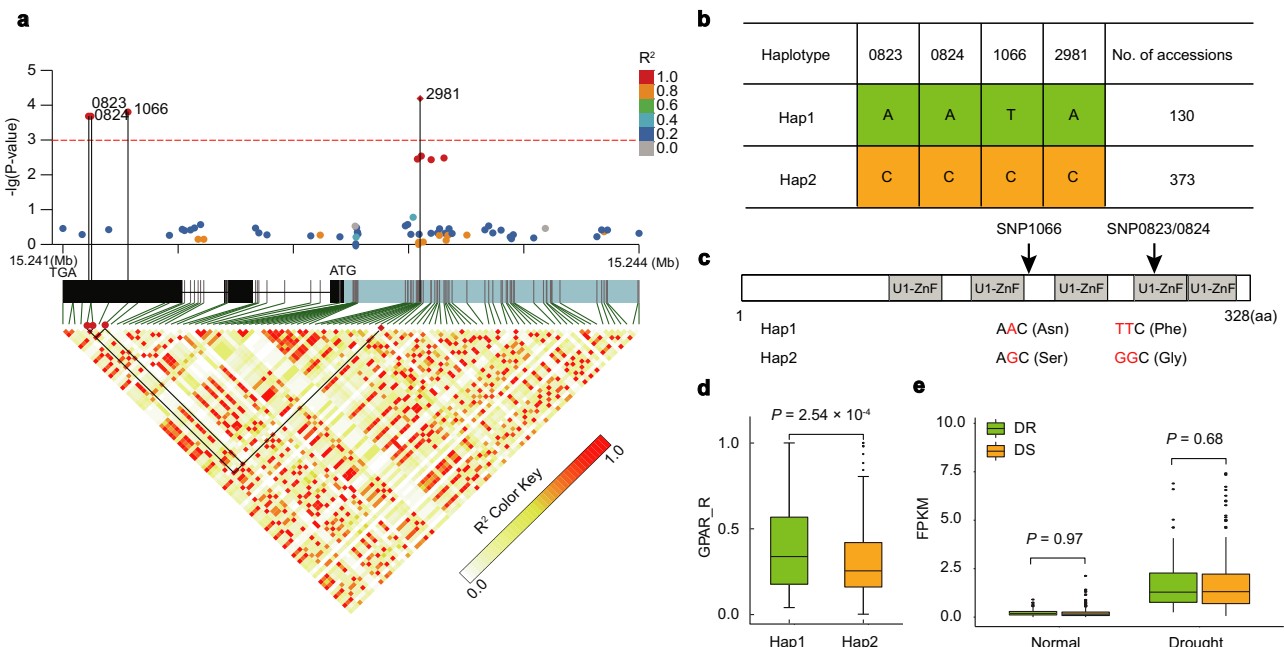

**Fig. 1 | Significant association of *DRG9* with relative green-projected area ratio (GPAR_R), a drought resistance indicator. a** *DRG9*-based association mapping and pairwise LD analysis. Dots represent SNPs. The lead SNP is highlighted by diamond. The SNPs showing strong LD with the lead SNP are connected to the pairwise LD diagram with solid lines and highlighted with black lines. *P* values are calculated based on mixed linear model and the red dashed line indicates the significance threshold ($3.09 \times 10^{-3}$), which is determined by the Bonferroni test. **b** Major haplotypes of *DRG9*. The numbers 0823, 0824, 1066, and 2981 indicate the positions of the significantly associated SNP in *DRG9*. **c** SNPs (SNP 1066 A > G, SNP 0823 T > G, SNP 0824 T > G) in the coding sequence causing two amino acid substitutions (Asn to Ser and Phe to Gly, respectively). **d** Comparison of GPAR_R between the Hap1 and Hap2 using the two-tailed *t*-tests ($n = 130$ and 373 rice accessions). **e** Comparison of *DRG9* expression between the drought-resistant (DR) Hap1 and drought-sensitive (DS) Hap2 haplotypes. Gene expression levels were determined among 71 DR accessions and 222 DS accessions under normal condition and drought stress. For the box plots in **d**, **e**, the center line indicates the median, the edges of the box represent the first and third quartiles, the whiskers extend to 1.5 times interquartile range from the box edges, and dots denote outliers. Statistical significance was determined using the two-tailed *t*-test. Source data are provided as a Source Data file.

(Supplementary Fig. 3a, b). Subsequently, we examined the subcellular localization of DRG9 in the root tips of 7-day-old rice seedlings. Under normal conditions, the YFP fusion protein signals were diffusely distributed in the cytoplasm. However, when rice seedlings were treated with mannitol to simulate drought stress, the YFP-fusion protein condensed into cytoplasmic granule-like structures (Fig. 3a). Cycloheximide (CHX) pretreatment can inhibit the formation of DRG9 granules (Fig. 3a), indicating that the DRG9 granules belong to mRNP complexes. Notably, the formation of DRG9-YFP granules was dynamic and reversible. During recovery from mannitol treatment, the number of DRG9 granules decreased gradually and returned to the pre-stress state (Fig. 3b). We further examined the dynamics of DRG9-containing granules using fluorescence recovery after photobleaching assay (FRAP), and found that the fluorescent intensity of DRG9-containing granules partially yet rapidly recovered within 30 s after photobleaching, indicating that these structures are dynamic *in planta* (Fig. 3c, d). To determine whether the observed reversible cytoplasmic granules were SGs or PBs, we co-transformed rice protoplasts with mCherry-DRG9[Nip] along with the SG marker Rbp47b-YFP or the PB marker DCP1-YFP. We observed that the mCherry-DRG9[Nip] signals colocalized with the Rbp47b-YFP signals, but not with the DCP1-YFP signals, as revealed by the analysis of signal collinearity using the Pearson correlation coefficient (PCC) (Fig. 3e, f). These results suggested that DRG9 condensed into SGs under drought stress.

**DRG9 binds to mRNAs of stress-related genes including OsNCED4 and protects the OsNCED4 mRNA from degradation**

DRG9 is predicted to contain five zinc finger domains (Fig. 1c). Human Znf346 is a DRG9 homolog and has been reported to have dsRNA binding activity[35], but similar dsRNA-binding protein has not been reported in plants. To examine whether DRG9 has the dsRNA binding activity, we performed EMSA assay using recombinant GST-DRG9[Nip] fusion proteins (Supplementary Fig. 4a) with the T57A RNA, which has been reported to form dsRNA structure[36], ssRNA, ssDNA and dsDNA probes, respectively. The EMSA results showed that DRG9 can bind to dsRNA probes in vitro (Supplementary Fig. 4b). To identify the target RNA of DRG9 in rice, we performed RNA-binding protein immunoprecipitation sequencing (RIP-seq) using drought-treated DRG9[Nip]–3×Flag transgenic rice seedlings (Supplementary Fig. 4c) and identified 1378 transcripts that were associated with DRG9 (Supplementary Data 3). These transcripts include *OsNCED4*, *OsANN1*, *OsSLAC1* and *OsCHS1*, which have been reported to be involved in plant abiotic stress responses (Supplementary Fig. 4d). Gene Ontology (GO) analysis revealed that DRG9-associated transcripts were enriched in plant stress responses (Supplementary Fig. 4e). Given that DRG9 was located into the SGs, which are crucial sites for mRNA storage and protection, we hypothesized that DRG9 may regulate the stability of its bound mRNA. To validate this hypothesis, we performed RNA-seq profiling of ZH11 and *drg9* mutants (*drg9-26* and *drg9-29*) under drought stress to identify differentially expressed genes (DEGs) (Supplementary Fig. 5a and Supplementary Data 4). Subsequently, we selected 479 DEGs identified from both mutant lines as reliable DEGs for downstream analyses (Supplementary Fig. 5b). Through the overlap of DEGs and the bound genes from RIP-seq, we identified 30 candidate genes that might be directly bound and regulated by DRG9 (Supplementary Fig. 5c). Among them, the expression of *OsNCED4* (*LOC_Os07g05940*) is significantly reduced in *drg9* mutants and *OsNCED4* has been reported to involve in regulating ABA biosynthesis in rice, making it a potent downstream component of *DRG9* in regulating drought resistance (Supplementary Fig. 5d). To test whether

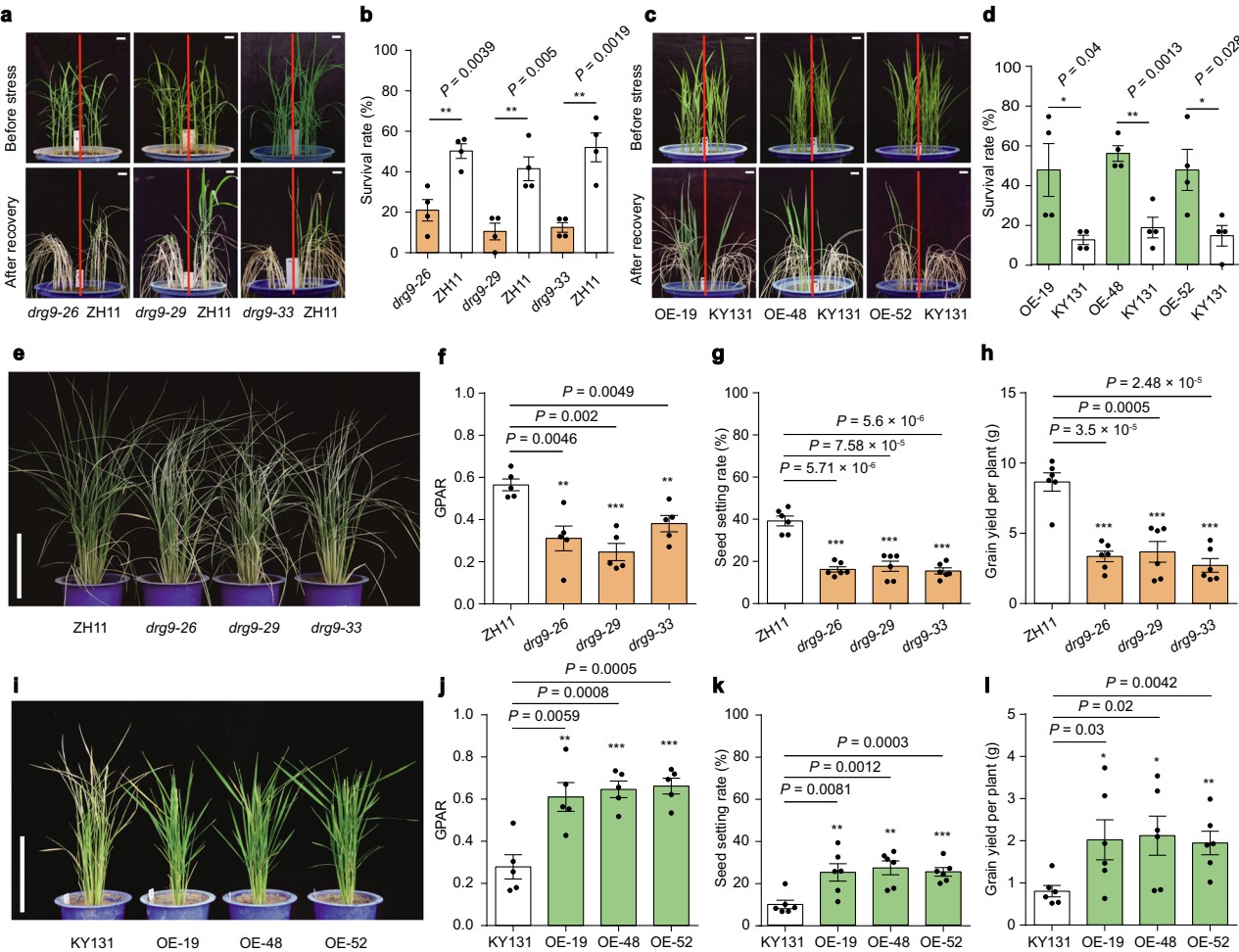

**Fig. 2 | Functional validation of *DRG9* in rice drought resistance. a** Drought resistance of *drg9* mutant compared to wild-type ZH11. Scale bar, 2 cm. **b** Survival rates of *drg9* mutant and ZH11 seedlings after re-watering. Data are means ± SEM ($n = 4$ biological replicates). **c** Drought resistance of *DRG9* OE lines compared to wild-type KY131. Scale bar, 2 cm. **d** Survival rates of *DRG9* OE lines and KY131 seedlings after re-watering. Data are means ± SEM ($n = 4$ biological replicates). **e** Growth performance of ZH11 and *drg9* mutant in the plot under drought stress. Scale bar, 20 cm. **f** GPAR of ZH11 and *drg9* mutant. Data are means ± SEM ($n = 5$ plants). **g** Seed-setting rate of ZH11 and *drg9* mutant. Data are means ± SEM

($n = 6$ plants). **h** Grain yield per plant of ZH11 and *drg9* mutant. Data are means ± SEM ($n = 6$ plants). **i** Growth performance of KY131 and *DRG9* OE lines in the plot under drought stress. Scale bar, 20 cm. **j** GPAR of KY131 and *DRG9* OE lines. Data are means ± SEM ($n = 5$ plants). **k** Seed-setting rate of KY131 and *DRG9* OE lines. Data are means ± SEM ($n = 6$ plants). **l** Grain yield per plant of KY131 and *DRG9* OE lines. Data are means ± SEM ($n = 6$ plants). Asterisks indicate statistical significance by two-tailed *t*-tests (*$P < 0.05$, **$P < 0.01$, ***$P < 0.001$). Source data are provided as a Source Data file.

the interactions between DRG9 and *OsNCED4* mRNAs occur in SGs, we isolated SGs using mannitol treated DRG9^Nip-YFP/*drg9* transgenic plants (Supplementary Fig. 6a) and used the SG fraction for further RIP assay. RIP-qPCR results showed that DRG9 can bind *OsNCED4* mRNAs in SGs (Supplementary Fig. 6b). Furthermore, we observed that the abundance of *OsNCED4* mRNA decreased in *drg9* mutant lines while increased in *DRG9* OE lines under both normal and drought stress conditions (Fig. 4a, b). To test whether *OsNCED4* mRNA stability is altered while bound by DRG9, we evaluated the *OsNCED4* mRNA stability in 7-day-old seedlings treated with cordycepin, a transcriptional inhibitor. We found that *OsNCED4* mRNA showed higher degradation rate in *drg9* mutants and lower degradation rate in *DRG9* OE lines compared to their respective controls (Fig. 4c, d), indicating that *DRG9* positively regulates the *OsNCED4* mRNA stability.

Considering the ability of DRG9 in binding dsRNA, we predicted the secondary structure of *OsNCED4* mRNA and found that its 3′-UTR (276 bases) can form double-stranded RNA structure (Fig. 4e). To verify whether DRG9 can bind 3′-UTR of *OsNCED4* in vitro, we performed an EMSA experiment, and confirmed that Cy5-labeled *OsNCED4* 3′-UTR

RNA probe was bound by GST-DRG9^Nip, but not by GST tag (Fig. 4f). To confirm that the 3′-UTR is required for the *OsNCED4* mRNA stability regulated by DRG9, we transiently expressed a firefly luciferase reporter gene tailed with (or without) *OsNCED4* 3′-UTR in rice protoplasts. These protoplasts were then transfected with either an empty vector (EV) or a DRG9^Nip expression vector. We found that the expression of DRG9^Nip increased mRNA level of the luciferase reporter gene with *OsNCED4* 3′-UTR, but not the level of the luciferase reporter gene without *OsNCED4* 3′-UTR (Fig. 4g). To investigate whether *OsNCED4* is responsible for drought resistance in rice, we generated *osnced4* mutant (Supplementary Fig. 7a), and observed that these mutant plants exhibited drought-sensitive phenotypes (Supplementary Fig. 7b, c). To test whether *OsNCED4* genetically acts downstream of *DRG9*, we generated *drg9/osnced4* double mutant. After drought stress, the survival rate of the double mutant resembled that of the *osnced4* single mutant (Fig. 4h, i), supporting that *OsNCED4* genetically acts downstream of *DRG9*. Together, these results suggested that DRG9 functions in regulating rice drought resistance by binding and promoting *OsNCED4* mRNAs stability.

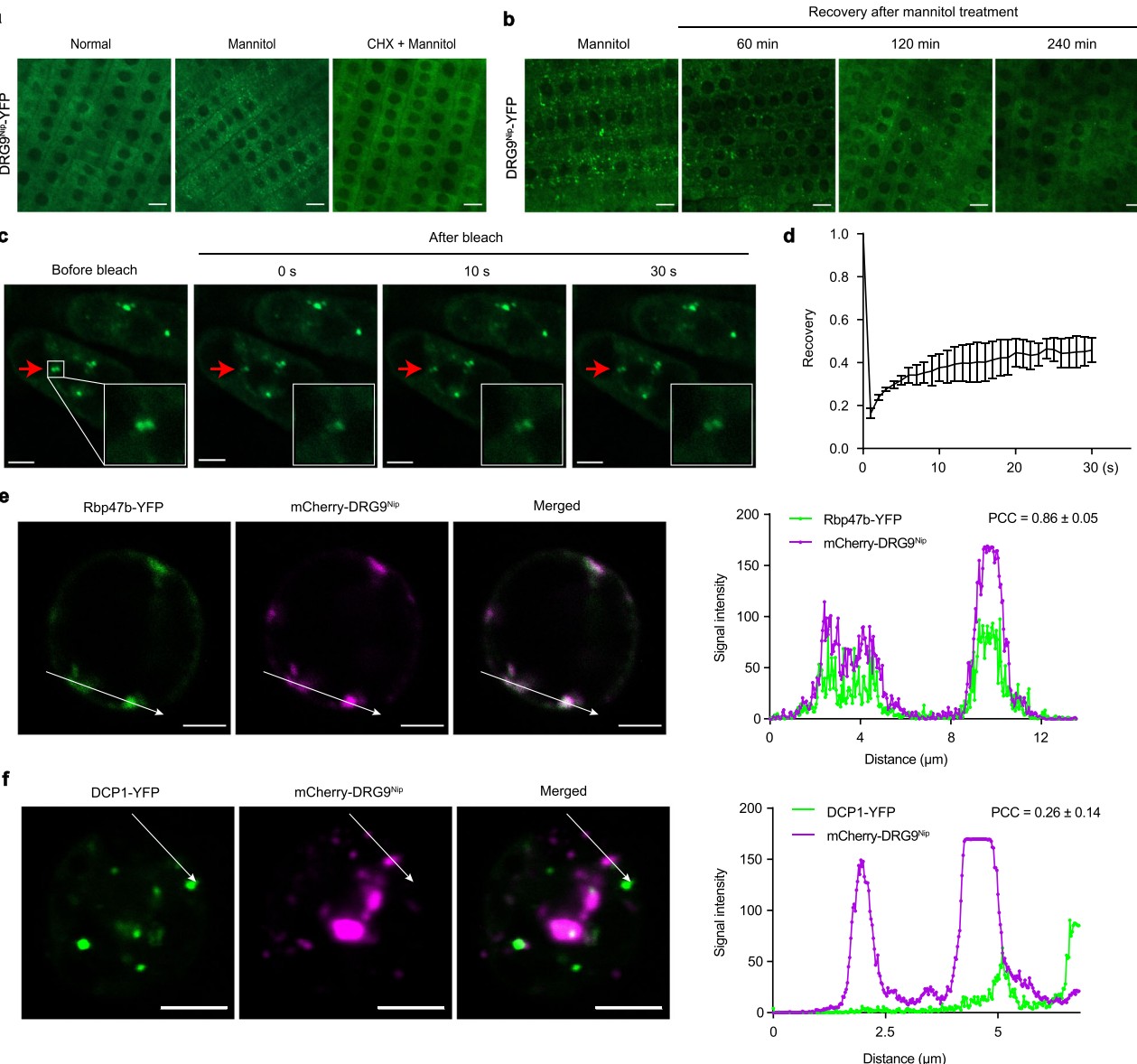

**Fig. 3 | DRG9 condensed into dynamic cytoplasmic granules under drought stress. a** Subcellular localization of rice root tips cells expressing DRG9^Nip-YFP under the control of the maize ubiquitin promoter under normal condition, mannitol treatment, and CHX treatment plus mannitol treatment, respectively. Scale bars, 10 μm. **b** Subcellular localization of DRG9^Nip-YFP during recovery. Scale bars, 10 μm. **c** FRAP of DRG9^Nip-YFP granules in the cytoplasm of root tip cells. Time 0 indicates the time of the photobleaching pulse. Scale bars, 5 μm. **d** Plot showing the time course of the recovery after photobleaching. Data are presented as the

mean ± SD (n = 3 independent experiments). **e, f** Representative fluorescence microscopy images, the signal intensity profiles and Pearson correlation coefficient (PCC) of rice protoplasts co-expressing Rbp47b-YFP or DCP1-YFP with DRG9^Nip-mCherry. Scale bars, 5 μm. The white arrows indicate the area of signal intensity profiles. n = 23/20 for PCC quantification respectively (mean ± SD was shown). In **a, b**, a representative experiment from three independent experiments is shown. Source data are provided as a Source Data file.

## A monocot-conserved α-helix is essential for DRG9 phase separation and function

Proteins that undergo liquid-liquid phase separation (LLPS) frequently encompass intrinsically disordered regions (IDRs). Using the IUPred algorithm[37], we predicted an IDR at the N-terminus of DRG9 spanning residues 1-99 (Fig. 5a). To determine whether the IDR itself is a driver for the DRG9 phase separation observed in vivo, we generated transgenic plants expressing DRG9ΔIDR-YFP, in which the IDR was removed in the *drg9* mutant background (Supplementary Fig. 8a), and found that DRG9ΔIDR-YFP lost the ability to form observable granules after mannitol treatment (Fig. 5b), suggesting that IDR of DRG9 is responsible for its LLPS. Subsequently, we expressed IDR-YFP of DRG9 in *E. coli* and purified the recombinant protein (Supplementary Fig. 8b). These proteins displayed phase separation and formed spherical

droplets under various conditions, including 5% PEG8000 and 10% dextran in vitro (Supplementary Fig. 8c), and can be disrupted by 1,6-hexanediol treatment (Supplementary Fig. 8d, e). Notably, under the 5% PEG8000 condition, IDR-YFP were even observed to form droplets at a protein concentration as low as 1.5 μM (Fig. 5c), and the IDR-YFP signal of the droplets recovered after photobleaching in the FRAP experiment (Fig. 5d, e). These results demonstrate that the IDR of DRG9 is responsible for DRG9 phase separation in vivo and in vitro. Amino acid composition analysis indicated that the IDR contains high portions of arginine (Arg) and lysine (Lys) residues with positive charges (Supplementary Fig. 8f). To investigate whether these amino acids play a crucial role in phase separation, we expressed and purified four variants of IDR-YFP proteins with Arg and Lys mutated to Ala to a different extent (named IDR M1 to IDR M4, respectively (Fig. 5f)),

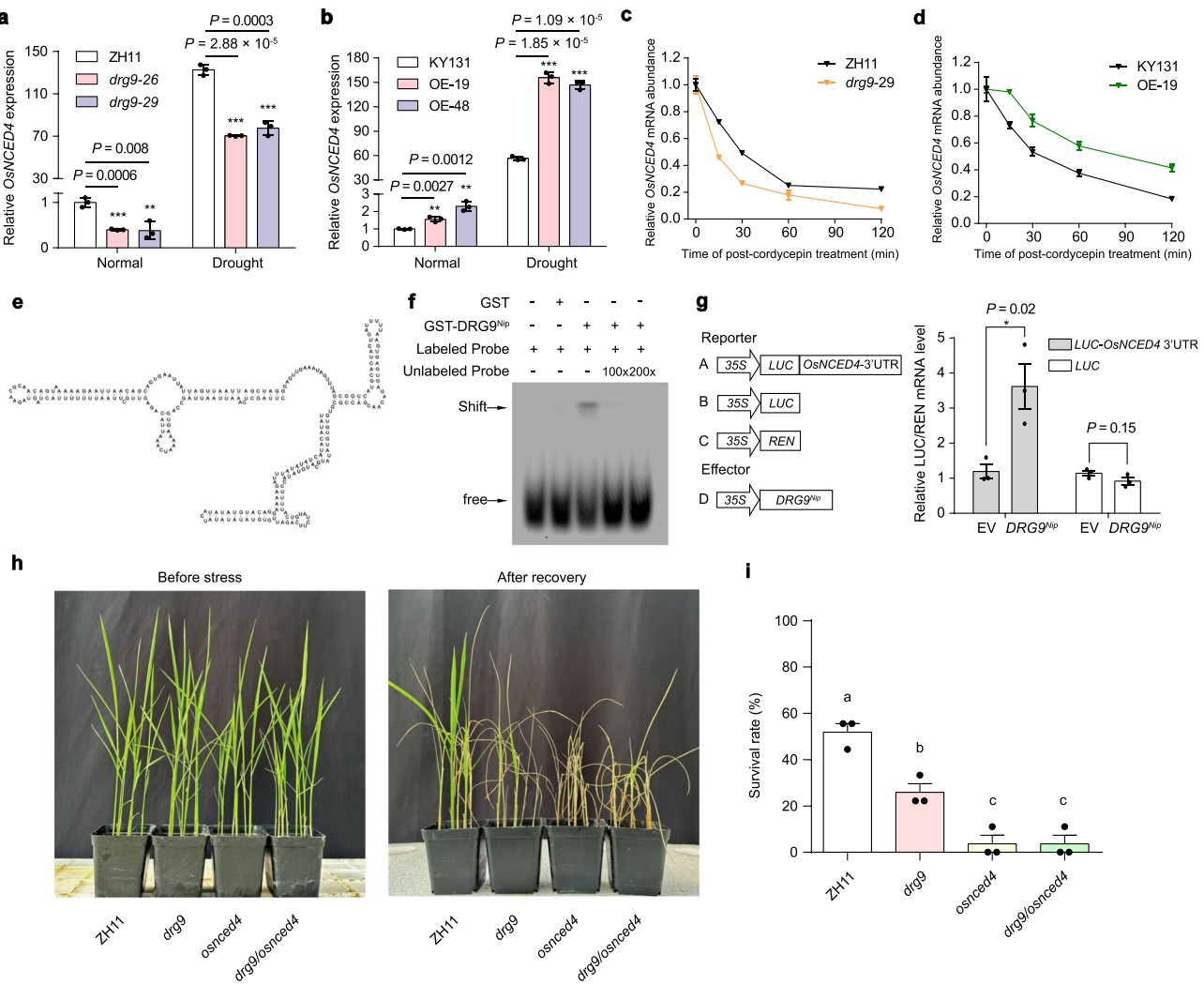

**Fig. 4 | DRG9 binds to and protects *OsNCED4* mRNA from degradation.**
**a** Relative mRNA levels of *OsNCED4* in ZH11 and *drg9* mutant under normal and drought stress. Data are means ± SEM (*n* = 3 biological replicates). **b** Relative mRNA levels of *OsNCED4* in KY131 and OE lines under normal and drought stress. Data are means ± SEM (*n* = 3 biological replicates). **c** Decay profiles of *OsNCED4* mRNAs in ZH11(WT) and *drg9* mutants (ZH11 background). Results from one representative replicate of two biological replicates are shown (each in three technical replicates). Data are means ± SD. **d** Decay profiles of *OsNCED4* mRNAs in KY131 (WT) and *DRG9* OE lines (KY131 background). Results from one representative replicate of two biological replicates are shown (each in three technical replicates). **e** Model for the secondary structure of the *OsNCED4* 3'UTR (total 276 bases). **f** EMSA results showing DRG9 binding to *OsNCED4* 3'-UTR dsRNA probes and competition by unlabeled probes. **g** The effect of DRG9^Nip on *LUC* mRNA stability. mRNA stability was measured by *LUC* mRNA level and normalized with *REN* mRNA level. Data are means ± SEM (*n* = 3 biological replicates). **h** Phenotype of ZH11, *drg9*, *osnced4*, and *drg9/osnced4* double mutant plants to drought stress. **i** Survival rates of ZH11, *drg9*, *osnced4*, and *drg9/osnced4* double mutant plants to drought stress. Data are means ± SEM (*n* = 3 biological replicates). Different letters indicate significant differences (*P* < 0.05, one-way ANOVA, Tukey's HSD test). Asterisks indicate statistical significance by two-tailed *t*-tests (*P* < 0.05, **P* < 0.01, ***P* < 0.001). In **f**, a representative experiment from three independent experiments is shown. Source data are provided as a Source Data file.

and found that the charged residues spanning residues 17-48 are crucial for LLPS as examined using recombinant proteins in vitro as well as tobacco system (Fig. 5g, h). To test whether these charged residues are necessary for the LLPS of DRG9 *in planta*, we generated transgenic plants expressing DRG9 M4-YFP and found that DRG9 M4-YFP lost the ability to form observable granules after mannitol treatment in rice root tip cells (Supplementary Fig. 8g). A closer examination in this region (residues 17-48) using AlphaFold structure prediction revealed that residues 17-48 could form α-helix (Fig. 5i). To further verify whether this α-helix is essential for driving phase separation, we inserted a proline residue into the center of the α-helix to break the α-helical structure. In tobacco cells, the proline-inserted IDR protein failed to form condensates after PEG6000 treatment (Supplementary Fig. 8h), suggesting that the intact α-helix is required for LLPS of DRG9. DRG9 homologs are present in monocot crops, with certain species containing more than one homolog (Supplementary Fig. 1c). Multiple

sequence alignment of DRG9 homologs revealed considerable variations in the zinc finger domains. Nonetheless, the region corresponding to the predicted α-helix remains highly conserved (Supplementary Fig. 8i), and AlphaFold structure prediction indicated the presence of the α-helix in the N-terminal IDR across all homologs (Supplementary Fig. 8j). We then expressed the IDR of DRG9 homologs in tobacco leaf cells and found that all of them exhibit disperse distribution under normal conditions but formed condensates following the PEG6000 treatment (Supplementary Fig. 8k), indicating the conserved phase separation ability of DRG9 homologs under osmotic stress. Finally, we examined whether the phase separation of DRG9 is essential for its function. We transiently expressed a firefly luciferase reporter gene tailed with (or without) the *OsNCED4* 3'-UTR in rice protoplasts. These protoplasts were subsequently transfected with empty vector, DRG9^Nip, or IDR-mutated DRG9^Nip (IDR M4) expression vector. The DRG9^Nip transfection increased mRNA level of the

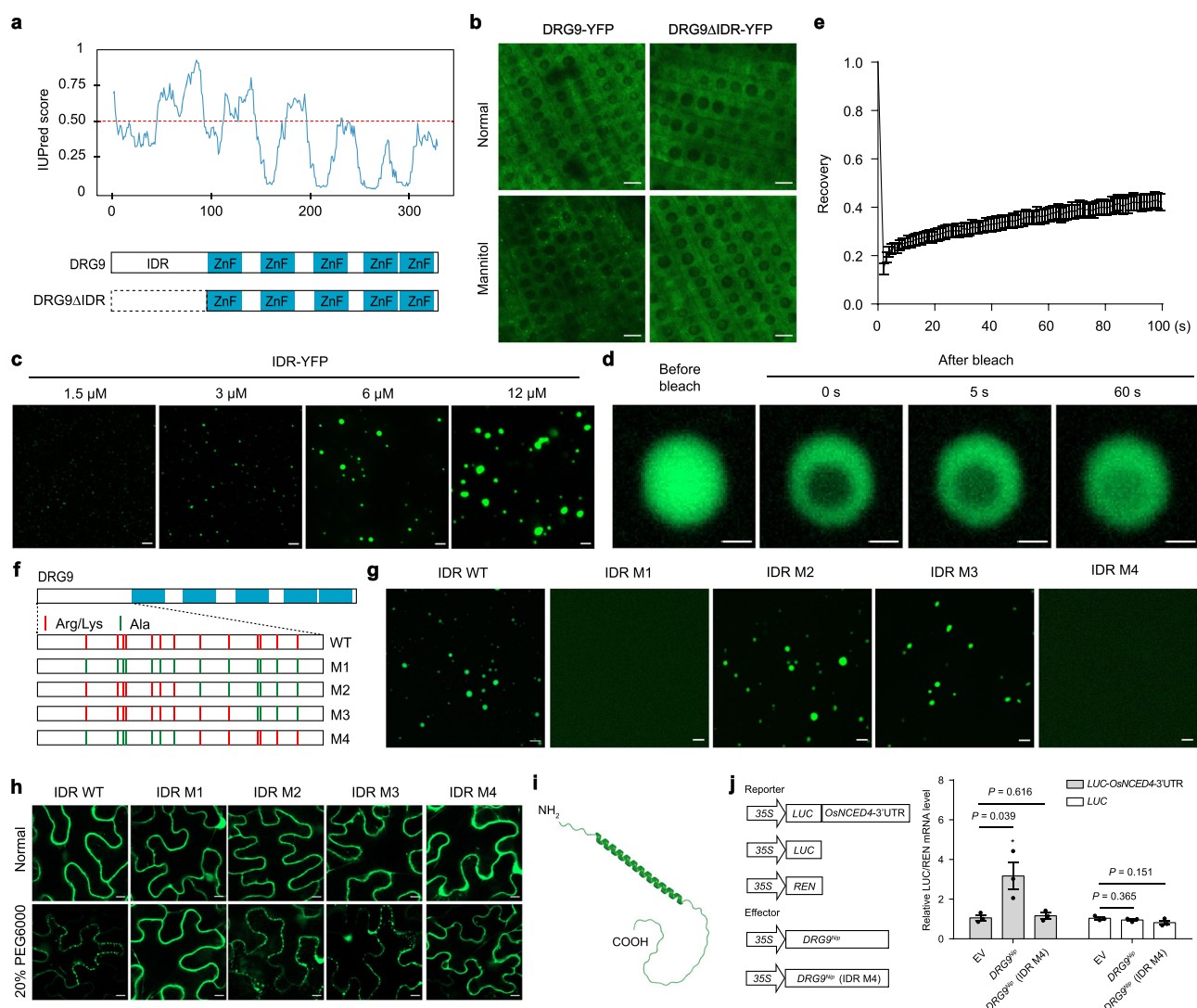

**Fig. 5 | An α-helix in the IDR is essential for DRG9 phase separation and function. a** Top, prediction of the disordered regions by the IUpred3 algorithm. Bottom, protein domain structures of DRG9 and its truncations. **b** Subcellular localization of rice root tips cells expressing DRG9-YFP or DRG9ΔIDR-YFP before and after mannitol treatment. Scale bars, 10 μm. **c** LLPS of purified IDR-YFP proteins in vitro. Different concentrations of IDR-YFP proteins were used. LLPS condition: 150 mM NaCl, pH 8.0 and 5% PEG8000. Scale bars, 5 μm. **d** FRAP of IDR-YFP droplets. Time 0 s indicates the time of the photobleaching pulse; scale bar, 1 μm. **e** Plot showing the time course of the recovery after photobleaching IDR-YFP droplets. Data are presented as the mean ± SD (n = 3 independent replicates). **f** Schematic presentation of the point mutations in DRG9 protein. Red lines indicate the

positions of Arg and Lys residues in the IDR, and green lines indicate Arg and Lys residues mutated to Ala. **g** In vitro phase separation assay of 12 μM indicated proteins in the presence of 5% PEG8000. Scale bars, 5 μm. **h** Subcellular localization of tobacco epidermal cells expressing IDR and IDR variants before and after PEG6000 treatment. Scale bars, 10 μm. **i** Structure of IDR as predicted by AlphaFold. **j** The effect of DRG9^Nip or DRG9^Nip (IDR M4) on *LUC* mRNA stability. The mRNA stability was measured by *LUC* mRNA level and normalized with *REN* mRNA level Data are means ± SEM (n = 3 biological replicates). Asterisks indicate statistical significance by two-tailed *t*-tests (*P < 0.05). In **b**, **c**, **g**, **h**, a representative experiment from three independent experiments is shown. Source data are provided as a Source Data file.

luciferase reporter gene, but DRG9^Nip (IDR M4), which couldn't undergo LLPS, did not increase mRNA level of the luciferase reporter gene (Fig. 5j). Collectively, these results indicated that phase separation of DRG9 was essential for its function.

### Natural variations in DRG9 affect its binding ability to OsNCED4 mRNA

We further explored whether natural variations in *DRG9* affect its protein functionality. Among the SNPs exhibiting strong LD with the lead SNP, three SNPs (SNP0823/T > G, SNP0824/T > G and SNP1066/A > G) were located in the coding region of *DRG9* (Supplementary Fig. 9a). Compared to the *DRG9*-DR alleles such as in rice Nipponbare (*DRG9^Nip*), SNP0823 and SNP0824 resulted in the substitution of Phe by Gly (F267G) and SNP1066 resulted in the substitution of Asn by Ser

(N186S) in the *DRG9*-DS alleles such as in rice Minghui 63 (*DRG9^MH63*) (Fig. 1c). We initially observed the subcellular localization of DRG9^Nip, DRG9^MH63, DRG9^Nip (N186S) and DRG9^Nip (F267G) in the rice root tips under normal and mannitol treatment, revealing no significant difference in the number of granules between different alleles after mannitol treatment (Supplementary Fig. 9b, c). Since the amino acid variations (N186S and F267G) were located in the zinc finger domain of DRG9 (Supplementary Fig. 9d), we speculated that this variation may affect the binding ability of DRG9 to its targets. To verify this hypothesis, we performed RIP experiment using transgenic rice of DRG9^Nip–3×Flag and DRG9^MH63–3×Flag. The RIP-qPCR result indicated that the binding ability to *OsNCED4* mRNA by DRG9^MH63 was significantly reduced compared to that by DRG9^Nip (Fig. 6a). Subsequently, we generated recombined proteins GST-DRG9^Nip and GST-DRG9^MH63 for RNA EMSA

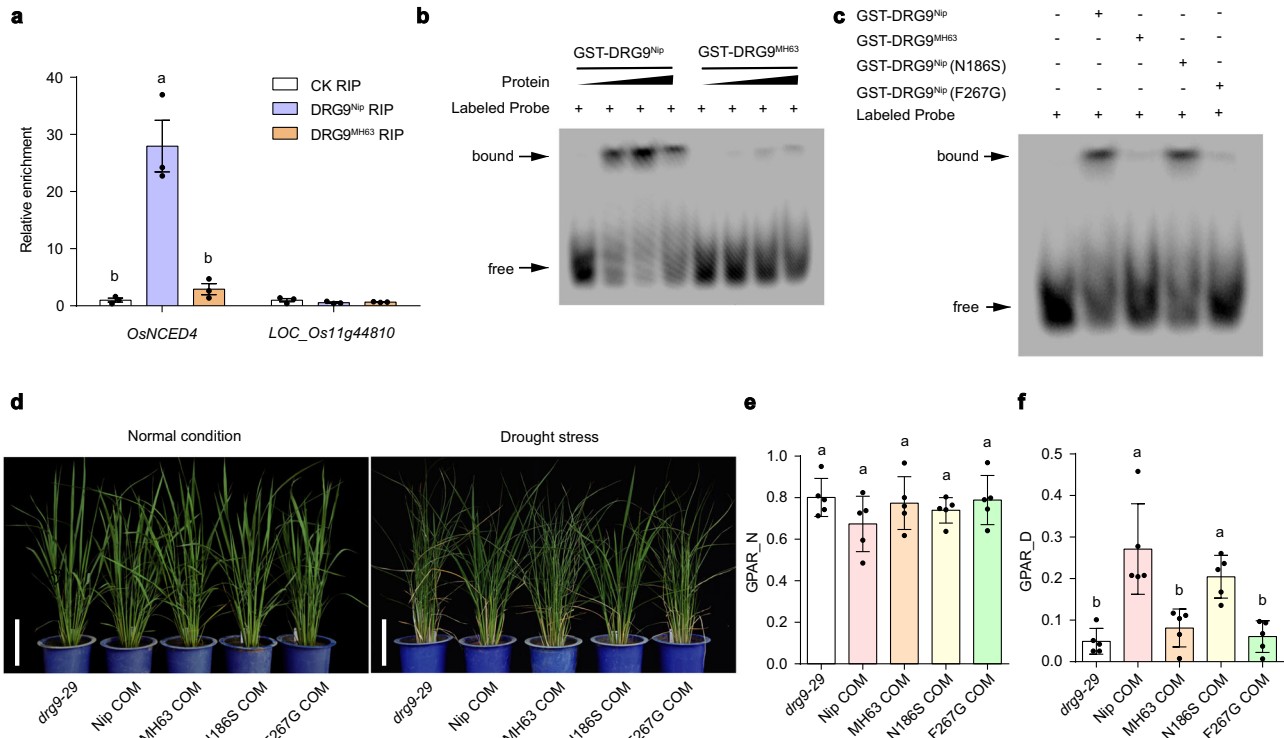

**Fig. 6 | Natural variation in DRG9 affects its binding activity to *OsNCED4* mRNA and drought resistance function. a** RIP-qPCR validation of the binding of DRG9[Nip] (F267) and DRG9[MH63] (G267) to *OsNCED4* mRNA. *LOC_Os11g44810*, a DRG9 non-target, was used as the negative control. Data are means ± SEM (*n* = 3 biological replicates). **b** EMSA results showing binding ability of GST-DRG9[Nip] and GST-DRG9[MH63] to *OsNCED4* 3'-UTR RNA probes. **c** EMSA results showing binding ability of GST-DRG9[Nip], GST-DRG9[MH63], GST-DRG9[Nip] (N186S), GST-DRG9[Nip] (F267G) to *OsNCED4* 3'-UTR RNA probes. **d** The performance of *drg9*, Nip COM, MH63 COM, Nip (N186S) COM and NIP (F267G) COM plants under normal and drought conditions. Scale bar, 20 cm. **e** GPAR of *drg9*, Nip COM, MH63 COM, Nip (N186S) COM and Nip (F267G) COM plants under normal conditions. Data are means ± SEM (*n* = 5 plants). **f** GPAR of *drg9*, Nip COM, MH63 COM, Nip (N186S) COM and Nip (F267G) COM plants under drought conditions. Data are means ± SEM (*n* = 5 plants). In **a**, **e**, **f**, different letters indicate significant differences (*P* < 0.05, one-way ANOVA, Tukey's HSD test). In **b**, **c**, a representative experiment from three independent experiments is shown. Source data are provided as a Source Data file.

experiments (Supplementary Fig. 9e). These experiments revealed that GST-DRG9[MH63] exhibited markedly weak, if any, binding ability to *OsNCED4* 3'-UTR RNA probes in contrast to the strong binding observed with GST-DRG9[Nip] (Fig. 6b). Further, we generated GST-DRG9[Nip] variant proteins, with mutations N186S and F267G, respectively (Supplementary Fig. 9e). RNA EMSA experiments indicated that the N186S mutation had no discernible impact on the binding affinity to *OsNCED4* 3'-UTR RNA, while the F267G mutation exhibited weak binding affinity to *OsNCED4* 3'-UTR RNA probes similar to GST-DRG9[MH63] (Fig. 6c). We also assessed the genetic effects of these variations by transforming the *drg9* mutant with *DRG9*[Nip], *DRG9*[MH63], *DRG9*[Nip] (N186S), and *DRG9*[Nip] (F267G) alleles, respectively, under the control of *DRG9*[Nip] promoter. Drought testing of these complementation lines revealed that both *DRG9*[Nip] and *DRG9*[Nip] (N186S) lines rescued the drought-sensitive phenotype of the *drg9* mutant, whereas *DRG9*[MH63] and *DRG9*[Nip] (F267G) did not (Fig. 6d, e). We further constructed a near-isogenic line (NIL), introducing the *DRG9*-DR allele from the rice germplasm WAB462 to Huanghuazhan (HHZ), a modern rice variety widely planted in the central and southern China. The resulting NIL[*DRG9*-DR] plants exhibited enhanced resistance to drought stress at the seedling stage compared to HHZ (Supplementary Fig. 10a, b). We also observed that the abundance of *OsNCED4* mRNA increased in NIL[*DRG9*-DR] seedlings compared to HHZ under drought stress conditions (Supplementary Fig. 10c). Because *OsNCED4* is a key enzyme in ABA biosynthesis, we further measured the ABA levels in the HHZ and NIL[*DRG9*-DR] seedlings and found the ABA level was significantly increased in the NIL[*DRG9*-DR] plants under drought stress conditions (Supplementary Fig. 10d). Collectively, these results indicate that the *OsNCED4* mRNA binding ability of the DRG9-DS type proteins is

weakened due to the F267G natural variation, resulting in a decrease in the *OsNCED4* mRNA abundance and ABA content, which explains the reduced drought resistance of *DRG9*-DS germplasms.

## Haplotype analysis and natural selection of DRG9 alleles

To further explore the natural variations in *DRG9*, we extended the haplotype analysis by using rice germplasms from RiceVarMap[38] and 3 K Rice Genomes Project[39]. According to the haplotype analysis of 14 SNPs in the *DRG9* gene, 7 major haplotypes were identified in the population (Supplementary Fig. 11a). However, only the two haplotypes, *DRG9*-DS (Hap1) and *DRG9*-DR (Hap2), were identified in the expanded population when using the three SNPs (SNP0823, SNP0824, and SNP1066). The *DRG9*-DS and *DRG9*-DR haplotypes included 5 (Hap1a-1e) and 2 (Hap2a-2b) sub-haplotypes, respectively (Supplementary Fig. 11a). Among the germplasm accessions, 96.8% of aromatic accessions contained the *DRG9*-DR allele, whereas only 5.2% of *aus* accessions and 0.8% of *indica* accessions contained *DRG9*-DR allele. The frequency of the *DRG9*-DR allele was variable in *japonica* accessions: 73.1% in temperate *japonica*, 93.6% in intermediate *japonica* and 99% in tropical *japonica* (Supplementary Fig. 11b and Supplementary Data 5). Subsequently, we analyzed the frequency of *DRG9*-DR allele in upland rice and lowland rice, and found that the proportion of *DRG9*-DR haplotype in upland rice (64.1%) was much higher than that in lowland rice (35.5%) (Fig. 7a and Supplementary Data 6). This led us to speculate whether the allele distribution of *DRG9* contributes to their environmental adaptation, especially in relation to the precipitation in different geographical locations. We then collected data of annual precipitation in China, and analyzed the annual precipitation with *DRG9*-DR genotype frequency of major rice-producing regions in

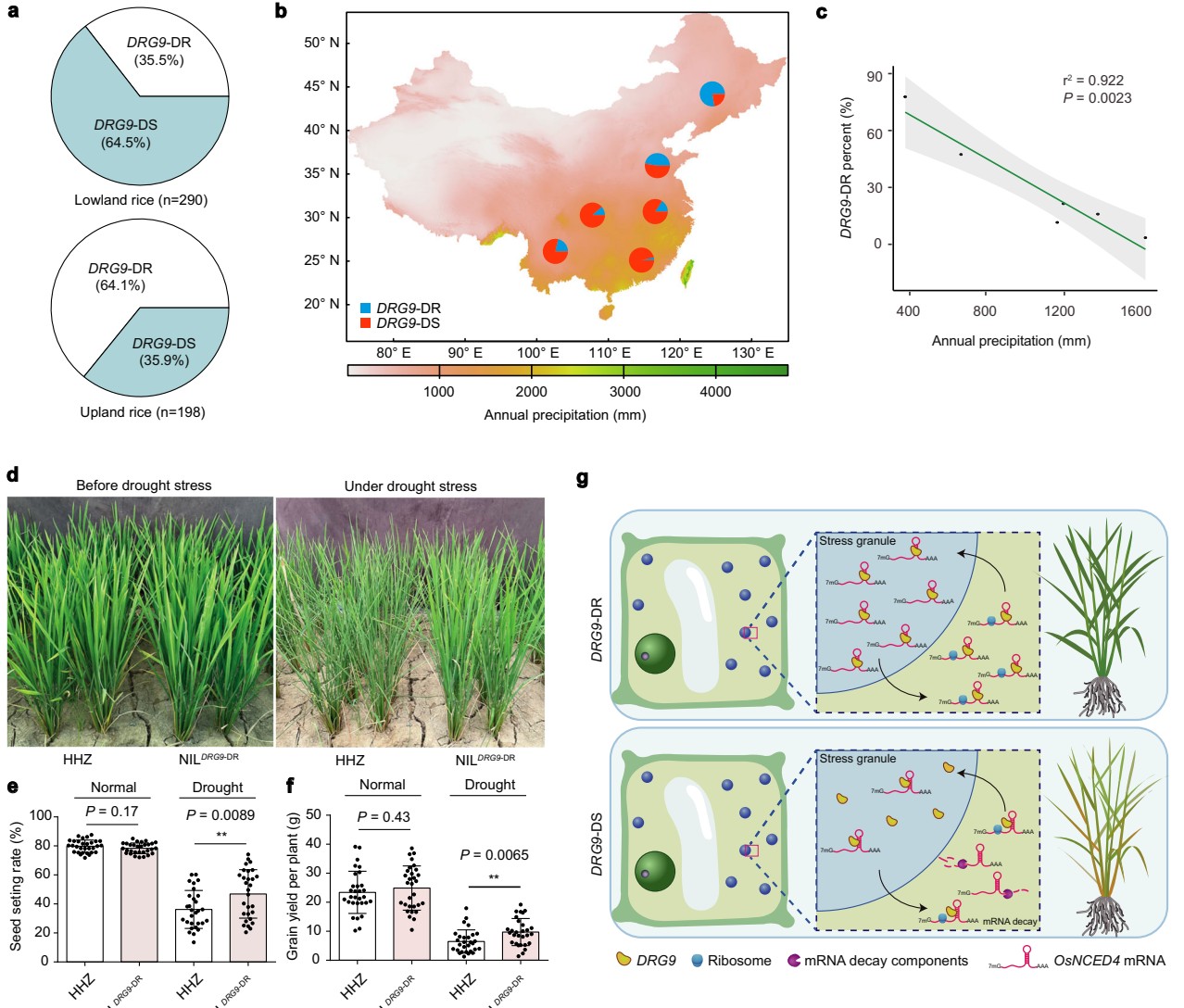

**Fig. 7 | Haplotype analysis and geographical distribution of *DRG9* allele. a** The proportions of *DRG9*-DR and *DRG9*-DS in upland rice and lowland rice. **b** The proportions of *DRG9*-DR and *DRG9*-DS haplotypes in the rice germplasms from major rice-planting regions in China. **c** The proportion of *DRG9*-DR is negatively correlated with annual precipitation with the Pearson correlation analysis (*n* = 6 rice-planting regions; *P* = 0.0023; the level of confidence interval was 95%). **d** The performance of HHZ and NIL^*DRG9*-DR plants before drought stress and under drought stress in the field. **e** Seed-setting rate of HHZ and NIL^*DRG9*-DR plants under normal and drought conditions. Data are means ± SEM (*n* = 30/30, 29/29 plants). **f** Grain yield per plant of HHZ and NIL^*DRG9*-DR plants under normal and drought conditions. Data are means ± SEM (*n* = 30/30, 29/29 plants). **g** A working model of *DRG9*-DR and *DRG9*-DS differentially regulating drought resistance. Asterisks indicate statistical significance by two-tailed *t*-tests (***P* < 0.01). Source data are provided as a Source Data file.

China (Supplementary Data 7). In Southeast China Hills with annual precipitation more than 1500 mm, *DRG9*-DS varieties widely exist (96.51%), while *DRG9*-DR accessions only account for 3.5%. In the Yun-Gui Plateau region, although the latitude was similar to that of the Southeast China Hills, the annual precipitation is markedly decreased, and the percentage of *DRG9*-DR accessions in this region is increased to 21.4%. Overall, there were more accessions carrying *DRG9*-DR in northern districts with low annual precipitation (Fig. 7b). A correlation test showed that *DRG9*-DR allele exhibited a negative correlation with the annual precipitation (r² = 0.922) (Fig. 7c and Supplementary Data 8). These results indicate that the *DRG9*-DR allele might have been selected for the adaptation of rice varieties to areas with low precipitation. To convince the genetic gains of the *DRG9*-DR allele in drought resistance breeding, we evaluated the drought resistance of NIL^*DRG9*-DR and HHZ with drought stress applied at panicle development stage in the field. During the process of stress development, the NIL^*DRG9*-DR plants showed much delayed leaf-rolling compared with HHZ

(Fig. 7d). Finally, NIL^*DRG9*-DR plants showed a significantly increase in seed-setting rate and grain yield per plant compared with that of HHZ after drought stress (Fig. 7e, f). Under well-irrigated conditions, the NIL^*DRG9*-DR and HHZ plants had similar performance for seed-setting rate and grain yield per plant (Fig. 7e, f). Moreover, no significant difference was detected between the NIL^*DRG9*-DR and HHZ in a number of agronomic traits (Supplementary Fig. 12a–e). Collectively, these results indicate that *DRG9*-DR holds great potential in drought resistance breeding in rice.

## Discussion
In this study, we cloned a drought-resistance gene *DRG9*, which encodes a dsRNA-binding protein, and found that DRG9 protects *OsNCED4* mRNAs at the post-transcriptional level via protein phase separation. DRG9 is a homolog of the human dsRNA-binding protein Znf346, which was reported to carry out its function in the nucleus[35]. However, we found that DRG9 was localized in the cytoplasm under

normal conditions and relocated to stress granules (SGs) under drought-stress conditions. SGs are assemblies of messenger ribonucleoproteins (mRNPs) that form from mRNAs stalled in translation initiation during stress responses[40]. Typically, these mRNAs are translationally competent and ready for translation during stress recovery. Distinct from its animal homologs, DRG9 contains an additional N-terminal intrinsically disordered region (IDR) that enables it to undergo LLPS and co-condensate into SGs in response to drought stress, and such phase separation is essential for DRG9's function in drought resistance. Additionally, we found that there is an α-helix embedded in the IDR that is indispensable for the condensate formation. Interestingly, the α-helix, which is highly conserved in DRG9 homologs of monocot plants, is also important for their LLPS behavior. Thus, we propose that the α-helix-mediated phase separation of the DRG9 homologs is likely a conserved mechanism in monocot plants in response to drought stress. Stress 'sensors' with LLPS capability have been reported recently in model plant Arabidopsis but not in crop plants[41,42]. The fact that DRG9 undergoes LLPS directly in response to drought stress in vivo and drought-mimic conditions in vitro makes it a potential drought stress 'sensor'. It would be interesting to see whether there are any natural variants of DRG9 or its homologs which possess altered LLPS behavior that may sense drought stress more swiftly, a property that could be applied in the process of genetic improvement of drought resistance in crops. In addition to SG localization, we observed that DRG9 condensates showed proximity to membrane surfaces in some cells (Fig. 3a). Similar localization pattern was found for Arabidopsis DCP1 condensates that was shown to interface with membranes[43]. Therefore, in future research, characterization of the potential membrane binding activity and the protein interactome of DRG9 may benefit the further understanding of the biological functions of DRG9.

Compared with the SG components recently identified in mammalian cells, little is known about the molecular composition of plant SG and the principles dictating mRNA selection for SG recruitment in plants. Our work revealed that DRG9 can bind to *OsNCED4* mRNAs and lead them into SGs for protection against degradation. Genetic experiments have shown that, the survival rate of *osnced4* mutants is less than that of *drg9* mutants, which might mean that the stabilization effect exerted by DRG9 is likely contributing moderately to drought resistance. This may be due to the *OsNCED4* expression is strongly induced by drought stress (Fig. 4a, b) and was reported to be activated by transcription factor, such as OsbZIP23 at the transcriptional level[16]. Besides being regulated at the transcriptional level, we found that DRG9 can stabilize the *OsNCED4* mRNA at the post-transcriptional level, jointly maintaining the high expression of *OsNCED4* under drought conditions, enhancing the biosynthesis of ABA, and thereby improving the drought resistance of rice. In addition to regulating mRNA stability, SG has also been reported to be involved in mRNA translation regulation. By analyzing transcriptome data, we found that many of the mRNA bound to DRG9 did not show significant changes. Therefore, we speculate that DRG9 may also play a certain role in regulating mRNA translation. Furthermore, we found that DRG9 is a dsRNA binding protein that favors RNA molecules with secondary structures. It is known that RNA secondary structures are closely related to their processing and function. For example, RNA duplex structures formed within mRNA molecules is crucial for their stability and translation[31,44], and RNA duplex structures formed between RNA molecules, such as mRNA-lncRNA and mRNA-mRNA, was reported to play an important role for the decay of these mRNAs in mammalian cells[45,46]. To identify RNA duplex structures, various computational modeling, evolutionary co-coupling analysis and chemical- or enzyme-based probing methods have been developed[47]. However, these duplex structures are mostly predicted or only confirmed in vitro. Therefore, identifying credible RNA secondary structures in a cellular context is crucial to the understanding their regulation and function.

In the *DRG9*-DR haplotype, a strengthened dsRNA binding affinity of DRG9 was due to the presence of a Phe at position 267. A previous study demonstrated that the zinc finger domain of Znf346 binds to the dsRNA backbone without sequence specificity, forming complexes with contacts between the dsRNA minor groove and the residues in the N-terminal β-strands, as well as between the dsRNA major groove and residues in the helix-kink-helix motif[36]. Protein structure prediction indicates that the 267th amino acid of DRG9 is located between the zinc finger domain (N-terminal β2-strands) and α1-helix (Supplementary Fig. 8c), with a predicted direct interaction with the dsRNA minor groove. Thus, it is likely that a Phe with bulky side chain at position 267 from the *DRG9*-DR haplotype may be more favorable for the interaction between DRG9 and dsRNA than a Gly that is found in the *DRG9*-DS haplotype.

The higher proportion of the *DRG9*-DR haplotype within the upland rice population raises intriguing questions about its potential exposure to natural selection in habitats with low precipitation. During rice domestication across diverse geographical locations, the water content in the field likely exhibited variability. In regions with low precipitation, the *DRG9*-DR allele may have been retained under selective pressure linked to drought stress. However, in the process of modern rice breeding, the absence of selective pressure might lead to the loss of the *DRG9*-DR allele in modern rice cultivars. Especially, *DRG9*-DR allele is generally lacking in the indica rice population. With the worsening global environment and frequent natural disasters such as drought in southern indica rice regions, utilizing the *DRG9*-DR allele to improve drought resistance in indica rice is of great significance for ensuring food security. Therefore, the *DRG9*-DR allele has promising value in breeding drought-resistant rice, which is highly relevant in the era of global climate change, when severe drought stress occurs more and more frequently.

## Methods

### Association analysis of DRG9 with drought resistance
In order to carry out association analysis for *DRG9*, a total of 62 SNPs in the genomic sequence containing 2.0 kb in the upstream of the promoter and spanning the 5'UTR to 3'UTR region of *DRG9* was analyzed, and the association with the GPAR_R of rice subjected to drought stress at the reproductive stage was calculated by TASSEL 5.0 using a standard mixed linear method (MLM, with MAF ≥ 0.05). LD and haplotype blocks were constructed using the LDBlockShow software[48].

### Plant materials
To construct *DRG9* knockout line using CRISPR-Cas9 technology, the 20 bp-specific single guide RNA target sequence of *DRG9* was designed and cloned into the TKC vector driven by OsU3 promoter[49]. The construct was then transformed into rice calli of the wild-type ZH11 (*O. sativa ssp. japonica*) through *Agrobacteria*-mediated transformation. To construct *OsNCED4* knockout line using CRISPR-Cas9 technology, two 20 bp-specific single guide RNA target sequences of *OsNCED4* was designed and cloned into the TKC vector driven by OsU3 and OsU6 promoter. The construct was then transformed into the wild-type ZH11. All mutations were confirmed by polymerase chain reaction (PCR) and sequencing. Cas9-free mutants that were homozygous for the mutations were then obtained and used for experiments.

To generate Ubq:DRG9^Nip constructs, coding sequences of DRG9^Nip were amplified from the cDNA of Nipponbare (Nip), and cloned into the pU1301U vector. The construct was then transformed into rice calli of the wild-type KY131 (*O. sativa ssp. japonica*) through *Agrobacteria*-mediated transformation. To generate Ubq:DRG9^Nip–3×Flag and Ubq:DRG9^MH63–3×Flag constructs, coding sequences of DRG9^Nip and DRG9^MH63 were cloned into pU1301U-3×Flag vector. The constructs were transformed into ZH11. To generate Ubq:DRG9-YFP constructs, coding sequences of DRG9^Nip, DRG9^NipΔIDR, DRG9^MH63, DRG9^Nip (N186S), DRG9^Nip (F267G) and

DRG9$^{Nip}$ M4 were cloned into pU1301U-YFP. The plasmids were transformed into $drg9$−29 mutant.

To construct the complement lines in $drg9$ background, genomic DNA fragments of DRG9$^{Nip}$, DRG9$^{MH63}$, DRG9$^{Nip}$ (N186S), and DRG9$^{Nip}$ (F267G), each driven by the DRG9$^{Nip}$ promoter (approximately 2.5 kb), were cloned into the pCAMBIA2300 vector. Each of these constructs was transformed into the $drg9$−29 mutant.

To develop the introgression line, an elite $indica$ variety HHZ, with the weak haplotype $DRG9$-DS was used as the recipient parent. The $japonica$ cultivar WAB462 with the strong haplotype $DRG9$-DR was used as the donor parent. WAB462 was crossed with HHZ to obtain $F_1$ seeds, which was backcrossed with HHZ for six generations to get $BC_6F_1$ plants and self-crossed for two generations to get $BC_6F_3$. During the backcross, $DRG9$-DR was selected by using the marker Id9N1524 (Supplementary Data 9). The introgression line carried an WAB462 genomic segment (approximately 300 kb) containing $DRG9$-DR, and the $BC_6F_4$ generation was used in this study.

### Drought resistance evaluation
For drought testing at the seedling stage, rice seeds were geminated on 1/2 Murashige & Skoog medium. At five days after germination, the seedlings were transplanted into pots filled with water-saturated soil. Each pot contained two genotypes (i.e., mutant line and WT) with an equal number ($n = 12$ plants per genotype) of plants for phenotypic observation, and at least three repeats were performed. Plants at the four-leaf stage were stressed with drought by stopping watering for 7-10 d, and re-watered until significant differences in seedling wilting were observed. After recovery, plants with green leaves and regenerating shoots were considered as survived plants and the survival rate (percentage of survived plants) was recorded. For drought stress testing at the reproductive stage, plants were planted in pots for drought treatment and trait measurement[34]. When rice plants grew to the reproductive stage, the plants designated for drought stress treatment were phenotyped by automatic phenotyping platform before stress treatment. Irrigation was then stopped to impose drought stress. The soil water content was monitored by weighing. When the soil water content decreased to 10%, the plants were watered once per day to maintain the soil water content at 10% for about 7 days. The drought-stressed plants were then phenotyped by automatic phenotyping platform again to collect phenotype data under drought stress. After collecting phenotypes, the plants were rewatered and subsequently collect yield traits.

### Subcellular localization
Microscopy images were acquired with a Zeiss LSM980 confocal microscope. To determine the subcellular localization of DRG9 in rice plants, we observed 7-day-old seedlings root tips under normal and mannitol treatment. For mannitol treatment, 7-day-old seedlings were transferred to 0.4 M mannitol and incubated for 30 min. For CHX treatment, 7-day-old seedlings were treated with 50 μM CHX and incubated for 30 min at room temperature, after which they were transferred to 0.4 M mannitol and incubated for 30 min. To determine the subcellular localization of DRG9 in rice protoplasts, the constructs 35 S:mCherry-DRG9$^{Nip}$ was co-transformed into rice protoplasts with 35 S:Rbp47b-YFP and 35 S:DCP1-YFP, respectively. Fluorescence was examined 16 h post-transformation under confocal microscopy. To determine the subcellular localization of the IDR of DRG9 and DRG9 homologs in $N. benthamiana$ epidermal leaf cells, the coding sequences of DRG9 IDR and its variants were cloned into pCAMBIA1301-35S-GFP vector. Vectors for protein expression were transformed into $Agrobacterium tumefaciens$ (GV3101). A colony was inoculated into liquid LB medium and cultured overnight. The cells were collected, resuspended in infiltration buffer (10 mM MES (pH 5.6), 10 mM MgCl$_2$ and 100 μM acetosyringone) and adjusted to an optical density at 600 nm (OD600) of 0.8. The suspended cells were infiltrated into

hydroponic tobacco leaves and incubated for 36−48 h before microscopy analyses. For PEG6000 treatment, hydroponic tobacco were transferred to 20% PEG6000 and incubated for 30 min.

### Fluorescence recovery after photobleaching (FRAP)
FRAP of DRG9-YFP condensates in rice root tip cells was performed on a Zeiss LSM980 confocal microscope using a ×40 objective. A region of a DRG9-YFP condensate was bleached using a 488-nm laser pulse (ten iterations, 100% intensity). Fluorescence recovery was recorded every 1 s for 60 s after bleaching. Images were acquired using ZEN software. FRAP of IDR-YFP droplets was performed on a Zeiss LSM980 confocal microscope using a ×40 objective. A small area of a droplet was bleached with a 488-nm laser pulse (fifteen iterations, 100% intensity). Recovery was recorded every 1 s for 100 s after bleaching. Images were acquired using ZEN software. The background values were subtracted from the fluorescence recovery values, and the resulting values were normalized by the first post-bleach time point and divided by the maximum point set maximum intensity as 1.

### Western blotting analysis
Protein samples were separated on SDS-PAGE gels and transferred to PVDF membranes. Antibodies against Flag (Mouse mAb; Sigma, F3165, Dilution 1;5000), GFP (Rabbit pAb; Abclonal, AE011, Dilution 1;5000) and Actin (Mouse mAb; Abclonal, AC009, Dilution 1;5000) were used as primary antibodies. After the primary antibody incubation, horseradish peroxidase-conjugated secondary antibodies goat anti-mouse (Abclonal, AS003, Dilution 1;10,000) and goat anti-rabbit (Abclonal, AS014, Dilution 1;10,000) were used for protein detection by chemiluminescence (Yeasen, 36208ES76).

### Analyses of RNA by RT-qPCR
Total RNA was isolated from rice leaves using $TransZol$ (TransGen, catalogue no. ET101-01-V2). Reverse transcription was carried out with a complementary DNA synthesis kit (TransGen, catalogue no. AE311). qPCR was performed using SYBR Green mix (TransGen, catalogue no. AQ101) on an Applied Biosystems 7500 Fast Real-Time PCR System. Quantitative variations were calculated by the relative quantification method ($2^{-\Delta\Delta CT}$). Ubiquitin was used as a reference gene in the RT-qPCR experiments. The RT-qPCR was performed three times independently. The primers used for the RT-qPCR are listed in the Supplementary Data 9.

### RIP-seq and RIP-qPCR
The RIP assay was performed as previously described[50]. Briefly, rice seedlings grown in pots filled with water-saturated soil. Plants at the four-leaf stage were stressed with drought by stopping watering until all leaves rolled, and the seedlings (3 g) were treated with 0.5% formaldehyde for 30 min under a vacuum for cross-linking, the reaction was quenched using 125 mM glycine solution for 5 min under a vacuum. The samples were washed three times with H$_2$O. After H$_2$O was removed by blotting the sample with a paper towel, the samples were fast-frozen and ground in liquid N$_2$. The resulting fine powder was suspended in 25 mL of lysis buffer (1.4 mM KH$_2$PO$_4$, 8 mM Na$_2$HPO$_4$, 140 mM NaCl, 2.7 mM KCl (pH 7.4), 0.5% Triton X-100, 1 mM PMSF, protease inhibitor cocktail (Roche), and 20 units/mL RNase inhibitor) at 4 °C for 15 min and centrifuged at 13,500 g for 15 min at 4 °C, and the supernatant was transferred to a fresh tube, 20 μL of each sample was saved as the input. The remaining supernatant was incubated with 50 μL of anti-Flag Magnetic Beads (SIGMA, M8823) overnight at 4 °C. Following immunoprecipitation, the beads were washed five times with 1.5 mL of lysis buffer. To elute the protein-RNA complexes, the beads were incubated with 50 μL of RIP elution buffer (100 mM Tris-HCl, pH 7.4, 100 mM NaCl, 10 mM EDTA, 1% SDS and 40 units/mL RNase inhibitor) at room temperature for 10 min with rotation. The beads were centrifuged at 1,500 g for 1 min and the supernatant was

saved. The elution was repeated with an additional 50 μL of RIP elution buffer and the two eluates were combined. The input and IP samples were treated with 1 μL of 20 mg/mL proteinase K at 65 °C for 1 h. RNA was isolated using *Trizol* reagent, The extracted RNA was eluted with 15 μL water. Library construction and sequencing were performed by Novogene Inc (Tianjin, China) with HiSeq-PE150 (Illumina Inc., San Diego, CA, USA) using the paired-end sequencing strategy. Raw reads were firstly subjected to quality control using FastQC (v.0.20.1). Then reads were mapped to the rice genome (Nipponbare MSU 7.0) using HISAT2 (v.2.2.1) with default parameters. The read numbers mapped to each gene were counted using featureCounts (v1.5.0). The differential expression analysis was performed using DESeq2 R package (v.3.32.1). We defined the enriched RNAs in DRG9 RIP versus control as the RNAs showing at least 4-fold enrichment ($log_2FC > 2$) and an adjusted *P*-value less than 0.05.

For RIP-qPCR, After RNA was eluted, the relative enrichment of each RNA was determined by RT-qPCR. Primers used for the RIP-qPCR are listed in Supplementary Data 9.

### RNA-sequencing analysis
Total RNA was isolated from rice booting leaves under drought stress using *TransZol* (TransGen, catalogue no. ET101-01-V2). Library construction and sequencing were performed by Novogene Inc (Tianjin, China) with HiSeq-PE150 (Illumina Inc., San Diego, CA, USA) using the paired-end sequencing strategy. Raw reads were firstly subjected to quality control using FastQC (v.0.20.1). Then clean reads were mapped to the rice genome (Nipponbare MSU 7.0) using HISAT2 (v.2.2.1) with default parameters. The read numbers mapped to each gene were counted using featureCounts (v1.5.0). The differential expression analysis was performed using DESeq2 R package (v1.30.1). DEGs were identified as those with at least 2-fold change ($| log_2FC | > 1$) and an adjusted *P*-value less than 0.05.

### Electrophoretic mobility shift assay
The construct for in vitro protein expression was cloned by inserting the coding sequence of DRG9$^{Nip}$, DRG9$^{MH63}$, DRG9$^{Nip}$ (N186S) and DRG9$^{Nip}$ (F267G) into the pGEX4t-1 vector. GST-tagged DRG9 was expressed in the *E. coli* BL21 (DE3) strain and purified with glutathione Sepharose 4B (Amersham Biosciences) according to the manufacturer's instructions. The oligonucleotide sequences used in this study are listed in Supplementary Data 9. The indicated RNA or DNA oligonucleotides were synthesized and labelled with FAM at the 5′ end. The T57A RNA were heated at 95°C for 5 mins and annealed by gradually cooling down to 25 °C. To generate dsDNA, an equal amount of the complementary ssDNA was mixed, heated to 95 °C for 5 min, and annealed by gradually cooling down to 25 °C. *OsNCED4* 3′-UTR was transcribed from PCR products containing the T7 promoter and *OsNCED4* 3′-UTR sequences using HiScribe™ T7 Quick High Yeild RNA Synthesis Kit (NEB, #E2050S) according to the manufacturer's instructions. Briefly, 1 μg template DNA, NTP Buffer Mix, T7 RNA Polymerase were mixed, the reaction was incubated overnight at 37 °C. To generate Cy5-labeled *OsNCED4* 3′-UTR, 3 μg *OsNCED4* 3′-UTR RNA was mixed with 1× T4 RNA ligase buffer, 40 units T4 RNA Ligase 1 (NEB), 40 units Rnase Inhibitor, 1 mM ATP, 10% DMSO, 33.3 μM pCp-Cy5 (Jera Bioscience) and 15% PEG8000, the reaction was incubated overnight at 16 °C. Labeled RNA was purified and eluted with 20 μL water, and heated at 95 °C for 5 min and annealed by gradually cooling down to 25 °C. The oligonucleotides were incubated with purified DRG9 protein in 20 μL of binding buffer (20 mM Hepes (pH 7.5), 20 mM KCl$_2$, 1 mM MgCl$_2$, 10 mM ZnCl$_2$, 5 mM DTT, 10% glycerol, 0.1% NP-40) at room temperature for 1 h. The resulting protein-substrate complexes were resolved on 6% non-denaturing polyacrylamide gels at 100 V for 45 min in 0.5× TBE buffer. After electrophoresis, the FAM signals were detected using FUJIFILM FLA-5100 and the Cy5 signals were detected using FUJIFILM FLA-9000.

### mRNA decay analysis
7-day-old ZH11, *drg9*, KY131, *DRG9* OE seedlings pre-treated in 0.2 M mannitol for 12 h. After mannitol treatment, the rice seedlings were treated with 1 mM of cordycepin (Solarbio) followed by vacuum infiltration for 15 min. After cordycepin treatment, the tissues were harvested at indicated time points. Two independent biological replicates were performed with approximately 3 seedlings per sample in each biological replicate. The total RNA was isolated and performed RT-qPCR analysis. Primers used for the RT-qPCR are listed in Supplementary Data 9.

### Isolation of stress granules
SGs were isolated as previously described[51]. In brief, 14-day-old 35 S:GFP and UBQ:DRG9$^{Nip}$-YFP transgenic seedlings were harvested and fixed as previously mentioned in RIP-qPCR. The frozen samples were pulverized with a precooled mortar and pestle in 10 mL of lysis buffer (50 mM Tris-HCl pH 7.4, 100 mM potassium acetate, 2 mM magnesium acetate, 0.5% NP-40, 0.5 mM DTT, 1 mM NaF, 1 mM Na$_3$VO$_4$, protease inhibitor cocktail (Roche), and 40 units/mL RNase inhibitor). The suspension was centrifuged at 4,000 g for 10 min at 4 °C and the pellet was resuspended in 4 mL of lysis buffer after removing the supernatant. About 200 μL of the suspension was saved as the input and the remaining suspension was centrifuged at 18,000 g for 10 min at 4 °C. The pellet was resuspended in 1.5 mL of lysis buffer and further centrifuged at 18,000 g for 10 min at 4 °C. The pellet was resuspended in 1 mL of lysis buffer and centrifuged at 850 g for 10 min at 4 °C. The supernatant was examined under confocal microscopy for successful isolation of SGs and used for RIP-qPCR as previously mentioned. Primers used for the RIP-qPCR are listed in Supplementary Data 9.

### Dual-luciferase reporter assay in protoplasts
The *OsNCED4* 3′-UTR and cauliflower mosaic virus 35 S terminator (35 S Ter, control) fragments were cloned into pDONR221 using the Gateway BP clonase II enzyme (Invitrogen) and were subsequently moved into p2FLGW7 vector with the Gateway LR clonase II enzyme (Invitrogen) to generate 35 S:LUC-*OsNCED4* 3′-UTR and 35 S:LUC, respectively. The related primers are listed in Supplementary Data 9. 14-day-old rice seedlings were prepared for protoplasts[52]. Empty vector or effectors (35 S:DRG9$^{Nip}$ or DRG9$^{Nip}$ M4) were cotransformed with reporters into rice protoplasts. RNA was isolated 16 h after transfection with TRIzol. The RNA samples were reverse transcribed and quantified using quantitative RT-qPCR. Each assay was performed three times independently in protoplasts.

### In vitro LLPS assay
To generate the constructs for in vitro protein expression, DRG9 IDR and the IDR variants CDS were clone into a modified pET28a-YFP expression vector with the YFP fused at the C terminus. All proteins were expressed in *E. coli* BL21 (DE3) cells overnight at 16 °C in the presence of 0.2 mM IPTG. Cells were collected and lysed in 20 mM Tris-HCl (pH 8.0), 150 mM NaCl. After removing the cell pellet, supernatant was loaded onto a Ni column. The bound protein was eluted in elution buffer (20 mM Tris-HCl (pH 8.0), 250 mM imidazole). Protein concentration was measured by KAIAO K5600 micro spectrophotometer. Protein samples were prepared in LLPS buffers with crowding effects as indicated, droplets were observed using a Zeiss LSM980 microscope equipped with a ×40 objective.

### Measurement of ABA content
ABA extraction and quantification was performed as previously described[53]. Briefly, 0.1 g of four-leaf stage seedlings were extracted twice with 750 mL of plant hormone extraction buffer (methanol: water: glacial acetic acid, 80:19:1, v/v/v) supplemented with 10 ng/mL d6-ABA internal standards. Quantification was performed in an AB SCIEX Triple Quad™ 5500 System.

## Haplotype analysis of *DRG9*

For investigating the natural variation of *DRG9* in germplasm, a toal of 4139 rice accessions were subjected to haplotype analysis. The SNPs of rice accessions were obtained from 3 K Rice Genome Project (https://snp-seek.irri.org/) and RiceVarMap (http://ricevarmap.ncpgr.cn/). A total of 14 SNPs were used for haplotype classification.

## Geographical distribution of rice varieties

For the geographical distribution of rice varieties, 652 varieties were projected on the major rice-producing regions in China according to their origin information, and the haplotypes of the 652 varieties were referred to the previous studies[38]. The average annual precipitation (1948-2016) of the major rice-producing regions in China were calculated according to the data provided by the National Earth System Science Data Center (http://www.geodata.cn/).

## Reporting summary

Further information on research design is available in the Nature Portfolio Reporting Summary linked to this article.

## Data availability

The RNA-seq data generated in this study have been deposited to the NCBI SRA database under accession code PRJNA1020189. The RIP-seq data generated in this study have been deposited to the NCBI SRA database under accession code PRJNA1063606. The orthologs of maize, wheat, and barley were downloaded from EnsemblPlants database [http://plants.ensembl.org/Oryza_sativa/Gene/Compara_Ortholog?db=core;g=Os09g0421700;r=9:15240448-15242482;t=Os09t0421700-01]. The natural variation of rice accessions were obtained from 3 K Rice Genome Project [https://snp-seek.irri.org/] and RiceVarMap [http://ricevarmap.ncpgr.cn/]. The predicted protein structures were downloaded from AlphaFold database [https://www.alphafold.ebi.ac.uk/] with a UniProt accession codes Q69P58 (DRG9), A0A3B6KIZ1 (TraesCS5A02G223100), A0A3B6LMI5 (TraesCS5B02G222200), A0A3B6MSJ5 (TraesCS5D02G230800), A0A804QAH2 (Zm00001eb313070), C0P658 (Zm00001eb313050), C5XCW1 (SORBI_3002G204700), K3ZUH7 (SETIT_030258mg), K3ZZ60 (SETIT_031892mg). Materials used in this study are available upon request. Source data are provided with this paper.

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

## Acknowledgements

This work was supported by the National Key Research and Development Program of China (2022YFF1001604), National Natural Science Foundation of China (31930080, 31821005), and the Foundation of Hubei Hongshan Laboratory (2021hszd011, 2021hskf003). The computations in this paper were run on the bioinformatics computing platform of the National Key Laboratory of Crop Genetic Improvement, Huazhong Agricultural University.

## Author contributions

L.X., H.W. and X.Lai. designed the study; H.W., T.Y., Z.G., Y.Y., Y.Z., Y.W., X.Li., B.L. performed all experiments. H.W., H.T., P.W. analyzed data. L.X., H.W., H.X. and X.Lai. wrote and revised the article.

## Competing interests

The authors declare no competing interests.
