## [Peer Review File · Nature Communications]

A double-stranded RNA binding protein enhances drought resistance via protein phase separation in riceReviewers' Comments:

Reviewer #1:

Remarks to the Author:

This is a very interesting study showing that condensates could be linked to important agronomical traits in crops. Hence, the work is significant and will attract broad interest. This is the first study, to my knowledge, that considers a mechanism for stabilizing RNAs through a condensate in a crop of high value (rice). Authors provide evidence for the involvement of a newly identified gene, DRG9, in drought tolerance of rice. They take up an interesting approach, starting from GWAS identification of the gene, all the way to biochemistry. They find that DRG9 presumably identifies and binds double-stranded RNAs related to ABA signaling. This function relates to condensates, such as SGs and PBs. Finally, they show that there is likely a link between certain DRG9 alleles with tolerance in populations of rice. I have the following remarks, comments, and suggestions that authors should take into consideration.

In general, the work is sound about physiology large number of genetics have been conducted. However, the mechanistic exploration falls short. Authors should try to further verify conclusions and claims, mainly by linking the LLPS to the stabilization effect they are proposing. Some interpretations need to be revised.

Major concerns/comments:

1. The links to the condensates SGs and PBs are weak. Is the RNA identified stabilized there or in the cytoplasm? PBs have recently been shown to stabilize RNAs in plants. Why do authors consider only PBs to this end? Furthermore, SGs have been suggested as translational points in animals. This can be discussed. It would also help if the authors provided evidence that the protected DRG9 is entering translation. They could also provide evidence that the localization of the targeted RNAs is in SGs. They can approach that using in situ detection (smFISH) or purifications of SGs followed by qRT-PCR.
2. PBs are constitutive structures in the cell. How do the authors reconcile the idea that they find localization of DRG9 in PBs?
3. Why the colocalization with PBs is not considered in the abstract and other places in the manuscript?
4. It is unclear to me whether there are DRG9 homologs in dicots. Given that there is a homologous protein in humans, I assume that there are proteins resembling DRG9 in dicots (and other plants).
5. There is a lot of jargon regarding GWAS, QTLs, LDs, and other terms, not explained for a broad audience.
6. Binding of single-stranded RNAs: In EMSA, authors use dsRNA but remains unclear if the protein can also bind to ssRNA. Furthermore, the numbers of identified RNAs as targets of DRG9 should be included in the manuscript, along with a proper analysis of GO enrichments, etc. Furthermore, it is unclear what is the overlap between DEGs and identified RNA targets of DRG9 (I had to look at the image to see numbers). Furthermore, are there other regions predicted to be dsRNA in NCED4?
7. How relevant is the suggested mechanism of drought tolerance: how much of the RNA is bound by DRG9 of the total pool in the cell?
8. It is unclear to me how the 5'-UTR/3'-UTR ratio links to mRNA stability. I assume that they mean that a 5/3 ratio can give an idea of the deadenylation that precedes decapping (reference there is missing). Yet, many RNAs are truncated on the 3'-UTR to localize in PBs. I suggest authors conduct a more comprehensive analysis of the mRNA level using cordycepin treatments and qRT-PCR to measure stabilities.
9. Localizations: judging from the images with tagged DRG9, it seems that DRG9 has an interesting association with membranes which is not discussed at all in the manuscript. A protein (SFH8) with a similar IDR was recently published by Liu et al., (Plos Biol, 2023) and shows binding on membranes.
10. Why only the IDR was used for in vitro assays? This dataset makes little sense to me. Authors

should express full-length proteins with or without IDRs. Otherwise, they can rely on the in vivo experiments. Yet, I agree that the dataset is a relevant control for the M4 truncation.

11. 1,6-hexanediol could have been used to test for LLPS in vitro and in vivo. Do the DRG9 granules dissolve through the application of cycloheximide?

12. Line 178: are the protoplasts stressed to make the SGs appear? For this image, calculations of colocalization should be provided (PPC and/or Manders). Do the DRG9 always colocalize with SGs and PBs and to what extent? It looks a bit weird that they can colocalize with both, almost 100%, considering that these are independent structures. Did the authors use proper controls for signal bleedthrough (between channels)?

13. The survival rate of *nced4* mutants is less than that of DRG9. That would mean that the stabilization effect exerted by DRG9 is likely contributing moderately to drought tolerance. Hence, this needs to be discussed. Furthermore, as aforementioned, the stabilization can take place in PBs.

14. I am just curious: did the authors find SNPs in the IDRs?

15. Fig. 7: the model is preliminary unless the authors manage to show a strong link between DRG9-SGs-RNA decay.

Minor concerns/comments:

Sentence lines 75-7 are not required. There are many species in which RNA processing in SGs/Pisare is not being studied.

Line 104: what is local LD? unclear to a broad audience.

Line 105: SNP? please, spell it out.

Line 137: the CRISPR nature should be described. Is it loss-of-function or partial? Were the protein levels verified somehow?

Line 142: OE: levels?

Fig. 4c: Please, provide the number of nucleotides in the manuscript

Line 350: please, provide the R in the text, as the correlation is strong.

Line 407: there are approaches in plants for in vivo determination of RNA structure (SHAPE etc.).

Hence, the statements in the following lines, are not entirely correct.

Methods: need to be more detailed. For example, the FRAP experiment is not adequately described.

Extended data Fig. 8: quantification of the different alleles with regards to localization to puncta is required. Do they differ in numbers?

Reviewer #2:

Remarks to the Author:

Wang et al. cloned a rice drought-resistant gene DRG9 using a GWAS method and investigated its likely mechanisms in conditioning drought resistance. The authors showed that DRG9 as a dsRNA-binding protein with an intrinsically disordered domain was relocated to stress granules (SG) upon osmotic stress and that DRG9 could bind NCED4 transcripts to slow down their turnover rates. Alleles of DRG9 that were able to relocate to SG or with a higher capacity to bind NCED4 exhibited heightened drought resistance. The authors also suggested that natural selection appears to favor drought resistant alleles of DRG9 in drought-prone regions and these germplasms could serve the breeding of new rice varieties. The manuscript was well organized and prepared.

Strength: A large number of experiments were performed to adequately address several important questions (e.g., phase separation, NCED4 binding, genetic complementation, natural variation), and the data presented are of high quality.

Weakness: The hypothesis that increased stability of NCED4 transcripts leads to drought resistance in DRG9-DR alleles was not further explored. Presumably, this would lead to a higher or sustained ABA

levels yet either ABA levels or related assays were not mentioned. The authors may have done these assays; probably the data might not be easily interpreted. This is not unexpected given the complexity of several issues involved (e.g., the function of SG is unknown, DRG9 targets are numerous including both positive and negative - such as OsP2C49 – regulators of drought resistance). Given the scope and focus of the current study, this reviewer considered this weakness to be minor and could be addressed in their future work.

Other minor comments:

Line 138, need to mention the nature of the mutations, which may affect the subsequent expression of DRG9 transgenes (e.g., line 161).

Line 191, for RIP-seq assays, please state whether (and how) the plants were treated with drought stress prior to the assays.

Line 212, a more direct method to measure mRNA degradation rates would be preferred.

Line 482, clarify 're-water until all leaves rolled'?

Figures:

Line 774, clarify 'Hap1=131, Hap2=374'

Fig. 2, the mutant plants in Fig. 2i appear to be shorter and with more tillers. Were these consistently seen?

Fig. 3a, need to mention in the legend (and text) the promoter used to drive the transgene DRG9nip-YFP, since the native gene is strongly induced by stress.

Fig. 5c legend, 'LLPS condition(s)': were both NaCl and PEG added as stated? Need to mention that the numbers in the fig. are protein concentrations.

Extended data Fig. 8 legend, change the title of the figure - the binding ability of these proteins was not shown in this figure.

Reviewer #3:

Remarks to the Author:

Drought is one of major abiotic stresses restricting rice growth and development, which usually leads to great loss of grain yield, resulting in huge economic losses worldwide. In recent years, the water demand for agriculture was growing constantly, while the dry weather happened more frequently. Therefore, it is urgent to identify drought resistance gene(s) for breeding rice varieties with high drought tolerance.

In this manuscript, the authors identified a drought resistance gene, DRG9, which encodes a double-stranded RNA (dsRNA) binding protein, contributes to drought resistance in rice by a genome-wide association study. The authors further revealed that DRG9 could increase drought resistance by promoting stability of OsNCED4 mRNA through protein phase separation. The authors also found that the drought-resistant DRG9 (DRG9-DR) allele showing a small proportion in cultivated rice, especially in indica rice (merely 0.8%). More importantly, the NILDRG9-DR generated by introducing the DRG9-DR allele into the cultivar Huanghuazhan (HHZ) that harbored the drought-sensitive DRG9 (DRG9-DS)

allele exhibited increased drought tolerance compared to HHZ. These works meant that the DRG9-DR allele had great potential for breeding drought-resistant rice. Overall, the topic of this manuscript is general interest for researchers understanding the mechanism of drought tolerance in rice and offers a viable strategy for breeding drought-resistant rice varieties. Even so, I still think that the authors should give attention to the following points and suggestions.

1. Among the 24 candidate genes, the authors considered ORF23 as the target gene, as it showed the greatest change in expression levels induced by drought. However, the other genes, such as ORF4, ORF6, and ORF10, the expression levels of which were at least 4 times under drought stress conditions compared to that under normal conditions. Variation in these drought-induced genes may also lead to changes in rice resistance to drought stress. So, how did the authors identify ORF23 as the target gene?
2. The authors proposed that DRG9 enhances drought resistance by enhancing the stability of OsNCED4 mRNA. As it was known, OsNCED4 encodes 9-cis-epoxycarotenoid dioxygenase (NCED), a key enzyme for ABA synthesis, which regulates the endogenous ABA content of plants. Therefore, to further support the authors' conclusion, I think it is necessary to clarify whether the variation of DRG9 resulted in the change of endogenous ABA content in rice plants.
3. The authors reported that the frequency of the DRG9-DR allele in aromatic accessions is 96.8%, in temperate japonica, intermediate japonica, and tropical japonica is 73.1%, 93.6%, and 99%, respectively, while it was merely 0.8% and 5.2% in indica and aus accessions, respectively. Out of curiosity, what's the distribution of the DRG9-DR allele in wild rice population? Whether the DRG9 gene was under selection during the process of domestication from wild rice to cultivated rice?
4. As described in line 305-307, the GST-DRG9MH63 exhibited markedly weaker binding ability to OsNCED4 3'-UTR compared to GST-DRG9Nip, which meant that GST-DRG9MH63 also had the minor ability of binding to its target OsNCED4 3'-UTR. Indeed, the Fig. 6b showed that the interaction between GST-DRG9MH63 and RNA probe was gradually strengthened with the increase of protein concentration. In line 310-312, why the F267G mutation resulted in a loss of binding affinity (rather than weaker binding ability), even though the N186S mutation had no discernible impact on the binding affinity to OsNCED4 3'-UTR?
5. The authors confirmed that DRG9 can bind OsNCED4, OsPAL1, OsCHS1 and Os4CL2 mRNA, and it was proved that DRG9 can promote the stability of OsNCED4 mRNA, but there is no evidence showed that DRG9 also promotes the stability of the other mRNAs. Therefore, it is recommended to modify the description in the discussion in line 397-398, otherwise, I think there should be more evidence to support this conclusion.
6. Please check the Fig.1a and Extended Data Fig.1a for inconsistent horizontal coordinates. In Fig.1a, DRG9 was located in the interval of 15.241-15.244 Mb, while in Extended Data Fig.1a, all 24 candidate genes were located in the interval of 15-15.2 Mb.
7. Fig. 6a-f is not mentioned in the main text, and it should be referenced in line 303, line 307, line 312, and line 317, respectively.
8. In Fig.7b, it is suggested to avoid using red and green simultaneously for the pie charts.
9. In line 887-888, the growth stage and tissues used for analyzing the expression levels of DRG9 should be described.
10. The abstract could be modified to be more concise.
11. In the part of discussion, I suggest that the value or advantage of the DRG9 gene in the application of rice breeding should be highlighted.

Reviewer #1 (Remarks to the Author):

This is a very interesting study showing that condensates could be linked to important agronomical traits in crops. Hence, the work is significant and will attract broad interest. This is the first study, to my knowledge, that considers a mechanism for stabilizing RNAs through a condensate in a crop of high value (rice). Authors provide evidence for the involvement of a newly identified gene, DRG9, in drought tolerance of rice. They take up an interesting approach, starting from GWAS identification of the gene, all the way to biochemistry. They find that DRG9 presumably identifies and binds double-stranded RNAs related to ABA signaling. This function relates to condensates, such as SGs and PBs. Finally, they show that there is likely a link between certain DRG9 alleles with tolerance in populations of rice. I have the following remarks, comments, and suggestions that authors should take into consideration.

In general, the work is sound about physiology large number of genetics have been conducted. However, the mechanistic exploration falls short. Authors should try to further verify conclusions and claims, mainly by linking the LLPS to the stabilization effect they are proposing. Some interpretations need to be revised.

Major concerns/comments:

1. The links to the condensates SGs and PBs are weak. Is the RNA identified stabilized there or in the cytoplasm? PBs have recently been shown to stabilize RNAs in plants. Why do authors consider only PBs to this end? Furthermore, SGs have been suggested as translational points in animals. This can be discussed. It would also help if the authors provided evidence that the protected DRG9 is entering translation. They could also provide evidence that the localization of the targetted RNAs is in SGs. They can approach that using in situ detection (smFISH) or purifications of SGs followed by qRT-PCR.

Response: Thanks for your kind and insightful comments and suggestions. We agree that although we mentioned that DRG9 can be localized in SGs, we lack sufficient evidence to support that DRG9 interacts with its binding mRNA (*OsNCED4* mRNA) in SGs. Indeed, in situ detection (smFISH) would be useful to understand the details of RNA localization. However, the corresponding detection system in rice has not been well established yet. Alternatively, according to the experimental methods in Arabidopsis, it seems feasible to isolate SG in rice. Thus, to investigate whether the interaction between DRG9 and *OsNCED4* mRNA in SG, we treated rice seedlings expressing DRG9-YFP with mannitol and isolated SGs. As shown in Extended Data Fig. 6a, DRG9-YFP were present in the SG-enriched fraction. As a control, there was no GFP protein present in the SG-enriched fraction isolated from 35S::GFP transgenic plants. Next, we conducted RIP-qPCR experiments using the isolated SG components.

As shown in Extended Data Fig.6b, we found that DRG9 can interact with *OsNCED4* mRNA in SGs. We have revised these results in the new version (line216-219).

Regarding DRG9's localization in PBs. In our initial manuscript, we found that DRG9 had some overlapping fluorescence signals with the PB marker DCP1, thus we propose that DRG9 may also localize in PB, however, we also did not have sufficient experimental data to support this. Now we have performed more explicit assays and analysis and conclude that DRG9 localize to SGs (please see details in question #2), and we added this data to our revised manuscript (Fig.3e,f). Indeed, SG has been reported as a venue for regulating translation. Considering that DRG9 binds to some mRNA and that we did not detect significant changes of these mRNA in the RNA-seq data, we think it is indeed very likely that DRG9 affects the function of these mRNA. The effect you mentioned on its translation may be one of the important ways, so we discussed it in the discussion (line 431-434). It would be interesting research in future work, we may further explore the impact of DRG9 on mRNA translation through ribosome sequencing.

2. PBs are constitutive structures in the cell. How do the authors reconcile the idea that they find localization of DRG9 in PBs?

Response: This question is related to your comments #12. Based on your suggestions on #12, we have conducted a co-localization analysis of DRG9 and the PB marker DCP1 in rice protoplasts again. As shown in Fig.3f, although we observed some colocalization of in fluorescence signals from the image, the proportion of such overlapping fluorescence signals was low and the overlapping fluorescence signals have different shapes (left panel). We have also conducted PCC analysis on this data as you have suggested, and found that the PCC was only 0.26 ± 0.14 (right panel). Therefore, we think that the observed overlapping fluorescence signals between two images (DRG9 and DCP1) was caused by random events. In contrast, DRG9 has much better overlapping signals with the SG marker Rbp47b (Fig.3e) and much higher PCC values (0.86 ± 0.05). Therefore, we conclude that DRG9 was mainly located in SGs, but not in PBs. We have revised these results in the new version (line182-185).

3. Why the colocalization with PBs is not considered in the abstract and other places in the manuscript?

Response: This question is addressed in concerns #1 and #2.

4. it is unclear to me whether there are DRG9 homologs in dicots. Given that there is a homologous protein in humans, I assume that there are proteins resembling DRG9 in dicots (and other plants).

Response: According to the analysis results of the Ensembl Plants database (http://plants.ensembl.org/Oryza_sativa/Gene/Comparative_Ortholog?db=core;g=Os09g0421700;r=9:15240448-15242482;t=Os09t0421700-01), we found that the DRG9 homologous gene only exists in many monocotyledonous plants and is missing in dicotyledonous plants, including the model plant Arabidopsis. Although we have found the DRG9 homologous gene Znf346 in animals, the amino acid sequence similarity between DRG9 and Znf346 is relatively low, with only partially similar sequences present in the zinc finger domain, and Znf346 also does not have an IDR sequence at its N-terminus.

5. There is a lot of jargon regarding GWAS, QTLs, LDs, and other terms, not explained for a broad audience.

Response: Thank you for your valuable suggestion, we have provided a detailed description of these jargons in line 41: quantitative trait locus (QTL), line 47: Genome-wide association studies (GWAS), line 48: linkage disequilibrium (LD) and line 104: single nucleotide polymorphism (SNP) to help the broad audience to understand these words.

6. Binding of single-stranded RNAs: In EMSA, authors use dsRNA but remains unclear if the protein can also bind to ssRNA. Furthermore, the numbers of identified RNAs as targets of DRG9 should be included in the manuscript, along with a proper analysis of GO enrichments, etc. Furthermore, it is unclear what is the overlap between DEGs and identified RNA targets of DRG9 (I had to look at the image to see numbers). Furthermore, are there other regions predicted to be dsRNA in *NCED4*?

Response: Thank you for your suggestion. In addition to the dsRNA probe, we now conducted the EMSA experiments of DRG9 using ssRNA, ssDNA, and dsDNA probes, respectively. As shown in Extended Data Fig.4b, we found that DRG9 did not exhibit binding activity to ssRNA, ssDNA, and dsDNA probes in vitro, but only to dsRNA. We have revised these results in the new version (line192-195).

Through RIP-seq, we identified 1378 mRNA that DRG9 can bind (revised in line 199). Based on your suggestion, we also conducted a GO analysis, and the results are revised in line 202-203 and shown in Extended Data Fig.4e.

As shown in Extended Data Fig.5c, by overlapping between the identified RNA targets of DRG9 and the RNA-seq DEGs, we identified a total of 30 genes that may directly regulated by DRG9, including *OsNCED4*.

In addition to the 3'UTR sequence of *OsNCED4* mRNA, we also predicted the secondary structure of the 5'UTR sequence. However, as shown in the following figure, the predicted results indicate that the 5'UTR sequence does not form much RNA secondary structures like the 3'UTR sequence do.

7. How relevant is the suggested mechanism of drought tolerance: how much of the RNA is bound by DRG9 of the total pool in the cell?

Response: Through GO analysis of 1378 genes identified by RIP-seq, we found that many pathways related to stress responses were enriched (Extended Data Fig.4e), including *OsNCED4*, *OsANN1*, *OsSLAC1* and *OsCHS1* which have been reported to be involved in plant abiotic stress responses (Extended Data Fig.4d). In this work, through genetic and phenotypic analysis we found that *OsNCED4* is the key downstream target of DRG9 that contribute to its positive role in drought resistance.

8. It is unclear to me how the 5-UTR/3-UTR ratio links to mRNA stability. I assume that they mean that a 5/3 ratio can give an idea of the deadenylation that precedes decapping (reference there is missing). Yet, many RNAs are truncated on the 3'-UTR to localize in PBs. I suggest authors conduct a more comprehensive analysis of the mRNA level using cordycepin treatments and qRT-PCR to measure stabilities.

Response: Thank you for your suggestion. In our initial manuscript, we referred to Zhou et al., 2017 (Loss of function of a rice TPR-domain RNA-binding protein confers broad-spectrum disease resistance. PNAS, 2017) and Zhang et al., 2022 (APIP5 functions as a transcription factor and an RNA-binding protein to modulate cell death and immunity in rice. Nucleic Acids Research, 2022), using mRNA 5-UTR/3-UTR ratio to represent mRNA stability. We also feel that this result is not intuitive enough, so based on your suggestion, we conducted an analysis of the *OsNCED4* mRNA level using cordycepin treatments and RT-qPCR to measure *OsNCED4* mRNA stabilities. As shown in Figure 4c/d, we found that *OsNCED4* mRNA showed higher degradation rate in *drg9* mutants compared with ZH11 WT plants and lower degradation rate in *DRG9* OE lines compared with KY131 WT plants. We have added these data to our revised version of the manuscript (line 222-227).

9. Localizations: judging from the images with tagged DRG9, it seems that DRG9 has an interesting association with membranes which is not discussed at all in the

manuscript. A protein (SFH8) with a similar IDR was recently published by Liu et al., (Plos Biol, 2023) and shows binding on membranes.

Response: Indeed, in some of the images the tagged DRG9 IDR proteins seemed to be associated with membrane, such as the ones in the tobacco system (Fig.5h).

In tobacco epidermal cells, the compression of vacuoles causes the cytoplasm to be located at the cell edge. Similar results have also been observed in some non-membrane localized proteins in the tobacco system (following figures), such as the expression of Rbp47b in the tobacco system (Phenolic acid-induced phase separation and translation inhibition mediate plant interspecific competition. *Nature Plants*, 2023 Extended Data Fig. 5d), expression of AtALKBH9B in the tobacco system (m6A RNA demethylase AtALKBH9B promotes mobilization of a heat-activated long terminal repeat retrotransposon in Arabidopsis. *Science Advances*, 2023 Fig. 4A), and expression of SEUSS IDR1 in the tobacco system (Condensation of SEUSS promotes hyperosmotic stress tolerance in Arabidopsis. *Nature Chemical Biology*, 2022 Fig. 2b).

In our study, in addition to the tobacco system, we also have DRG9 localization data in other systems. As shown in Figure 3a/b/c and Figure 3e, we observed the subcellular localization of DRG9 in cytoplasm of the rice root tip cells, and in cytoplasm of the rice protoplasts. In both cases, we did not observe DRG9's association with membrane. We had also read the SFH8 literature you recommended, we learned the experimental method of calculating colocalization (PCC) from the text, which has been very helpful for us to improve the manuscript.

10. Why only the IDR was used for in vitro assays? This dataset makes little sense to me. Authors should express full-length proteins with or without IDRs. Otherwise, they can rely on the in vivo experiments. Yet, I agree that the dataset is a relevant control for the M4 truncation.

Response: Because the recombinant 6×His-DRG9-YFP full-length protein is largely insoluble, we are unable to purify full-length protein for in vitro assays, so we used a fragment spanning the IDR (6×His-IDR-YFP) for *in vitro* LLPS assays. Based on your suggestion, in order to express sufficient soluble full-length DRG9 protein *in vitro*, we produced recombinant full-length DRG9 and DRG9ΔHelix fused with a solubility promoting tag (MBP tag), followed by a tobacco etch virus protease (TEV) cleavage site and a GFP tag. As shown in following figure, After TEV addition to cleave off MBP, DRG9-GFP formed puncta-like precipitate, and DRG9ΔHelix-GFP protein

exhibit soluble and disperse. These results indicate that the α -helix in IDR is necessary for DRG9 phase separation *in vitro*.

In addition, we also conducted *in vivo* experiments in rice. In our initial manuscript, as shown in Figure 5a/b, DRG9 Δ IDR lost the ability to form observable granules after mannitol treatment. Based on your suggestion, we also produced rice plants expressing DRG9 M4-YFP, as shown in Extended Data Fig.7g, DRG9 M4-YFP also lost the ability to form observable granules after mannitol treatment *in vivo*. We have revised these results in the new version (line274-278).

11. 1,6-hexanediol could have been used to test for LLPS in vitro and in vivo. Do the DRG9 granules dissolve through the application of cycloheximide?

Response: Thank you for your suggestion. We did this experiment as you have suggested. As shown in Extended Data Fig.7d/e, addition of 30% 1,6-hexanediol to a 1:1 volume can dissolve droplets. As a control, H₂O has no effect. We have revised these results in the new version (line262-263).

As shown in Figure 3a, pretreatment of rice seedlings with CHX, inhibited the formation of DRG9 granules after mannitol treatment. We have revised these results in the new version (line171-172).

12. Line 178: are the protoplasts stressed to make the SGs appear? For this image, calculations of colocalization should be provided (PPC and/or Manders). Do the DRG9 always colocalize with SGs and PBs and to what extent? It looks a bit weird that they can colocalize with both, almost 100%, considering that these are independent structures. Did the authors use proper controls for signal bleedthrough (between channels)?

Response: Thank you for your suggestion. During the cultivation of rice protoplasts, we think that these cells have been subjected to osmotic stress due to the presence of 0.6M mannitol in the WI buffer used for overnight cultivation. Thus, without additional treatment, we have observed the formation of cytoplasmic granules in SG's marker Rbp47b and DRG9. Unfortunately, although we conducted colocalization analysis using the rice protoplast system, we were unable to observe a state in which cells were not subjected to stress stimulation in rice protoplasts.

Based on your suggestion on SG and PB localization issues, we have performed additional experiments, and the results are shown and discussed in #1 and #2. For YFP fluorescent channel, Excitation wavelength: 508nm, Emission wavelength: 524nm, Detection wavelength: 524-564nm, Gain: 700V. For mCherry fluorescent channel, Excitation wavelength: 558nm, Emission wavelength: 583nm, Detection wavelength: 584-624nm, Gain: 700V.

13. The survival rate of nced4 mutants is less than that of DRG9. That would mean that the stabilization effect exerted by DRG9 is likely contributing moderately to drought tolerance. Hence, this needs to be discussed. Furthermore, as aforementioned, the stabilization can take place in PBs.

Response: The survival rate of *osnced4* mutants is less than that of *drg9* indeed merits more discussion. It could be that *OsNCED4* is not only regulated by DRG9 but by other factors as well, the mRNA level of *OsNCED4* in *drg9* mutant was partially affected, and the *osnced4* mutant leads to complete loss of its function, thus *osnced4* mutant shows more severe drought phenotype. We have added the discussion in lines 422-429.

14. I am just curious: did the authors find SNPs in the IDRs?

Response: In the rice population we studied, some varieties had some SNPs in the IDR. Since these SNPs are not significantly associated with drought resistance trait, and these SNPs are not located at the positions encoding α -helix (17-48 aa) in IDR, so we did not study further.

15. Fig. 7: the model is preliminary unless the authors manage to show a strong link between DRG9-SGs-RNA decay.

Response: Thank you for your suggestion. We hope that the additional experimental evidence described above can further prove the link between DRG9-SGs-*OsNCED4* mRNA decay in our model.

Minor concerns/comments:

Sentence lines 75-7 are not required. There are many species in which RNA processing in SGs/Pisare is not being studied.

Response: Thank you for your suggestion. We have removed this sentence in the revised manuscript.

Line 104: what is local LD? unclear to a broad audience.

Response: We have revised and provided a detailed description of this term in line 103.

Line 105: SNP? please, spell out.

Response: We have revised and provided a detailed description of this term in line 104.

Line 137: the CRISPR nature should be described. Is it loss-of-function or partial? Were the protein levels verified somehow?

Response: Through sequencing, we found that gene editing caused 1bp insertion in the second exon of DRG9, leading to frame-shift mutations. We have revised these results in the new version (line 136-138). Due to the lack of corresponding antibodies for DRG9,

we did not validate it at the protein level. As shown in following figure, in the transcriptome sequencing results, we found that the expression level of DRG9 decreased in drg9-26 and drg9-29 mutants. This is likely NMD (nonsense-mediated mRNA decay) mediated mRNA degradation. This may indirectly prove that the DRG9 mutant terminated translation prematurely due to a frameshift mutation.

Line 142: OE": levels?

Response: As shown in Extended Data Fig. 2b, the DRG9 expression level of the three independent OE lines are now added. We have revised these results in the new version (line142-143).

Fig. 4c: Please, provide the number of nucleotides in the manuscript

Response: We have revised the number of nucleotides in the text (line 229) and Fig.4e legend.

Line 350: please, provide the R in the text, as the correlation is strong.

Response: We have provided the R value in the text (line 377).

Line 407: there are approaches in plants for in vivo determination of RNA structure (SHAPE etc.). Hence, the statements in the following lines, are not entirely correct.

Response: Thank you for your suggestion. We revised the description of “development of a method to identify credible RNA secondary structures in cellular context is crucial to the understanding their regulation and function” to “identifying credible RNA secondary structures in cellular context is crucial to the understanding their regulation and function” in line 444-445.

Methods: need to be more detailed. For example, the FRAP experiment is not adequately described.

Response: Thank you for your suggestion. We have improved the description of the method section where it was not detailed enough, including FRAP experiment (line 549-559), RIP assay (line 578-584), DEG screening criteria in RNA-seq (line 621-622), Probe preparation in EMSA (line 628-643), mRNA decay analysis using cordycepin (line 651-657) and Isolation of stress granules (line 659-672).

Extended data Fig. 8: quantification of the different alleles with regards to localization to puncta is required. Do they differ in numbers?

Response: In order to quantify the data, we collected and analyzed images of plants with different alleles after mannitol treatment. As shown in Extend Data Fig8.b/c, there is no significant difference in the number of puncta between different alleles. We have revised these results in the new version (line 312-314).

Reviewer #2 (Remarks to the Author):

Wang et al. cloned a rice drought-resistant gene DRG9 using a GWAS method and investigated its likely mechanisms in conditioning drought resistance. The authors showed that DRG9 as a dsRNA-binding protein with an intrinsically disordered domain was relocated to stress granules (SG) upon osmotic stress and that DRG9 could bind NCED4 transcripts to slow down their turnover rates. Alleles of DRG9 that were able to relocate to SG or with a higher capacity to bind NCED4 exhibited heightened drought resistance. The authors also suggested that natural selection appears to favor drought resistant alleles of DRG9 in drought-prone regions and these germplasms could serve the breeding of new rice varieties. The manuscript was well organized and prepared.

Strength: A large number of experiments were performed to adequately address several important questions (e.g., phase separation, NCED4 binding, genetic complementation, natural variation), and the data presented are of high quality.

Weakness: The hypothesis that increased stability of NCED4 transcripts leads to drought resistance in DRG9-DR alleles was not further explored. Presumably, this would lead to a higher or sustained ABA levels yet either ABA levels or related assays were not mentioned. The authors may have done these assays; probably the data might not be easily interpreted. This is not unexpected given the complexity of several issues involved (e.g., the function of SG is unknown, DRG9 targets are numerous including both positive and negative - such as OsPP2C49 – regulators of drought resistance). Given the scope and focus of the current study, this reviewer considered this weakness to be minor and could be addressed in their future work.

Response: Thanks for your kind and insightful comments and suggestions. To verify the hypothesis that increased stability of *OsNCED4* transcripts leads to drought resistance in DRG9-DR alleles, we detected the transcription levels of *OsNCED4* in HHZ and NIL^{DRG9-DR} seedlings under normal and drought conditions. As shown in Extend Data Fig9.c, NIL^{DRG9-DR} has a higher transcription level of *OsNCED4* than the control HHZ under drought conditions. We also detected the ABA levels in HHZ and NIL^{DRG9-DR} seedlings under normal and drought conditions. As shown in Extend Data Fig9.d, NIL^{DRG9-DR} has a higher ABA level than the control HHZ under drought conditions. We have revised these results in the new version (line 334-344).

Other minor comments:

Line 138, need to mention the nature of the mutations, which may affect the subsequent expression of *DRG9* transgenes (e.g., line 161).

Response: Thank you for your suggestion. This information has now been added to the revised version of the manuscript (line 137).

Line 191, for RIP-seq assays, please state whether (and how) the plants were treated with drought stress prior to the assays.

Response: We have described the drought treatment conditions in line 197 and the methods section (line 578-580), respectively.

Line 212, a more direct method to measure mRNA degradation rates would be preferred.

Response: Thank you for your suggestion. We conduct an analysis of the *OsNCED4* mRNA level using cordycepin (a transcription inhibitor) treatments and RT-qPCR to measure *OsNCED4* mRNA stabilities. As shown in Figure 4c/d, we found that *OsNCED4* mRNA showed higher degradation rate in *drg9* mutants and lower degradation rate in *DRG9* OE lines compared to their respective controls. We have revised these results in the new version (line 222-227).

Line 482, clarify ‘re-water until all leaves rolled’?

Response: We apologize for the inappropriate description of the experiment. We have revised this description in line 521-522: re-watered until significant differences in seedling wilting were observed.

Figures:

Line 774, clarify ‘Hap1=131, Hap2=374’

Response: We have revised this description in the Fig.1d legend: n = 131 and 374 rice accessions.

Fig. 2, the mutant plants in Fig. 2i appear to be shorter and with more tillers. Were these consistently seen?

Response: In the previous planting process, we did not find significant differences between transgenic plants and wild-type controls. For the plants in the drought experiment shown in the Fig.2i, we also obtained plant height data through the phenotype platform at that time. As shown in the following figure, we found no significant difference in plant height.

Fig. 3a, need to mention in the legend (and text) the promoter used to drive the transgene DRG9^{nip}-YFP, since the native gene is strongly induced by stress.

Response: Thank you for your suggestion. We have revised that the promoter used to drive the transgene DRG9^{Nip}-YFP, in the text (line 163) and the Fig.3a legend

Fig. 5c legend, 'LLPS condition(s)': were both NaCl and PEG added as stated? Need to mention that the numbers in the fig. are protein concentrations.

Response: All proteins of different concentrations were added with NaCl and PEG as stated. We have revised in Fig. 5c legend: Different concentrations of IDR-YFP proteins were used.

Extended data Fig. 8 legend, change the tile of the figure - the binding ability of these proteins was not shown in this figure.

Response: Thank you for your suggestion. We have revised the tile of Extended data Fig. 9 legend (previous Extended data Fig. 8 legend): Natural variations in DRG9 leads to changes in amino acids in zinc finger domain.

Reviewer #3 (Remarks to the Author):

Drought is one of major abiotic stresses restricting rice growth and development, which usually leads to great loss of grain yield, resulting in huge economic losses worldwide. In recent years, the water demand for agriculture was growing constantly, while the dry weather happened more frequently. Therefore, it is urgent to identify drought resistance gene(s) for breeding rice varieties with high drought tolerance.

In this manuscript, the authors identified a drought resistance gene, DRG9, which encodes a double-stranded RNA (dsRNA) binding protein, contributes to drought resistance in rice by a genome-wide association study. The authors further revealed that DRG9 could increase drought resistance by promoting stability of OsNCED4 mRNA through protein phase separation. The authors also found that the drought-resistant DRG9 (DRG9-DR) allele showing a small proportion in cultivated rice, especially in indica rice (merely 0.8%). More importantly, the NILDRG9-DR generated by introducing the DRG9-DR allele into the cultivar Huanghuazhan (HHZ) that harbored the drought-sensitive DRG9 (DRG9-DS) allele exhibited increased drought tolerance compared to HHZ. These works meant that the DRG9-DR allele had great potential for breeding drought-resistant rice. Overall, the topic of this manuscript is general interest for researchers understanding the mechanism of drought tolerance in rice and offers a viable strategy for breeding drought-resistant rice varieties. Even so, I still think that the authors should give attention to the following points and suggestions.

Response: we thank you for your positive remarks on our manuscript.

1. Among the 24 candidate genes, the authors considered ORF23 as the target gene, as it showed the greatest change in expression levels induced by drought. However, the other genes, such as ORF4, ORF6, and ORF10, the expression levels of which were at least 4 times under drought stress conditions compared to that under normal conditions. Variation in these drought-induced genes may also lead to changes in rice resistance to drought stress. So, how did the authors identify ORF23 as the target gene?

Response: In our preliminary research, based on the screening of gene expression and annotation information, we selected ORF4, ORF6, ORF16, ORF19, ORF22, ORF23 as candidate genes for gene knockout. After phenotype identification, however, we found that only ORF23 had a drought-sensitive phenotype.

2. The authors proposed that DRG9 enhances drought resistance by enhancing the stability of OsNCED4 mRNA. As it was known, OsNCED4 encodes 9-cis-epoxycarotenoid dioxygenase (NCED), a key enzyme for ABA synthesis, which regulates the endogenous ABA content of plants. Therefore, to further support the

authors' conclusion, I think it is necessary to clarify whether the variation of DRG9 resulted in the change of endogenous ABA content in rice plants.

Response: To verify whether the variation of *DRG9* resulted in the change of *OsNCED4* mRNA abundance and endogenous ABA content in rice plants. We detected the transcription levels of *OsNCED4* in HHZ and NIL^{DRG9-DR} seedlings under normal and drought conditions. As shown in Extend Data Fig9.c, NIL^{DRG9-DR} has a higher transcription level of *OsNCED4* than the control HHZ under drought conditions. We also detected the endogenous ABA content in HHZ and NIL^{DRG9-DR} seedlings under normal and drought conditions. As shown in Extend Data Fig9.d, NIL^{DRG9-DR} has a higher ABA level than the control HHZ under drought conditions. We have revised these results in the new version (line 334-344).

3. The authors reported that the frequency of the DRG9-DR allele in aromatic accessions is 96.8%, in temperate japonica, intermediate japonica, and tropical japonica is 73.1%, 93.6%, and 99%, respectively, while it was merely 0.8% and 5.2% in indica and aus accessions, respectively. Out of curiosity, what's the distribution of the DRG9-DR allele in wild rice population? Whether the DRG9 gene was under selection during the process of domestication from wild rice to cultivated rice?

Response: Based on the wild rice data published by Jing et al., 2023 (Multiple domestications of Asian rice. *Nature Plants*, 2023), we analyzed the distribution of the *DRG9-DR* allele in wild rice population. As shown in the following figure, we found that the *DRG9-DR* allele was completely absent in Niv1 and Niv2, while the *DRG9-DR* allele was 23.5% in Ruf1 and 1.4% in Ruf2. This indicates that the *DRG9-DR* allele was retained during the process of domestication from *O. rufipogon* to *japonica*, which may have been selected. Due to the domestication process of *indica*, its main genome is derived from *O. nivara*, so the *DRG9-DS* allele is fixed, and a small number of *DRG9-DR* alleles in *Aus* (5.2%) may originate from inter-subspecies introgression.

4. As described in line305-307, the GST-DRG9MH63 exhibited markedly weaker binding ability to OsNCED4 3'-UTR compared to GST-DRG9Nip, which meant that GST-DRG9MH63 also had the minor ability of binding to its target OsNCED4 3'-UTR. Indeed, the Fig. 6b showed that the interaction between GST-DRG9MH63 and RNA probe was gradually strengthened with the increase of protein concentration. In line 310-312, why the F267G mutation resulted in a loss of binding affinity (rather than weaker binding ability), even though the N186S mutation had no discernible impact on the binding affinity to OsNCED4 3'-UTR?

Response: Indeed, it seems that F267G mutation resulted in weaker binding ability rather than a loss of binding affinity. To further verify this, we have now repeated all the EMSA experiment using freshly synthesized probes of RNA and freshly purified proteins (fig.4f, fig.6b, c). As shown in Figure 6c, compared to GST-DRG9 NIP and GST-DRG9 N185S, GST-DRG9 F267G indeed exhibit weak binding strength to probes similar to that of GST-DRG9 MH63. We apologize for incorrect description of the experimental results in our previous manuscript. Therefore, in line 327-329, we modify it to: the F267G mutation exhibited weak binding affinity to OsNCED4 3'-UTR RNA probes similar to that of GST-DRG9 MH63.

5. The authors confirmed that DRG9 can bind OsNCED4, OsPAL1, OsCHS1 and Os4CL2 mRNA, and it was proved that DRG9 can promote the stability of OsNCED4 mRNA, but there is no evidence showed that DRG9 also promotes the stability of the other mRNAs. Therefore, it is recommended to modify the description in the discussion in line 397-398, otherwise, I think there should be more evidence to support this conclusion.

Response: Thank you for your suggestion. Since we did not conduct further analysis on other mRNAs than *OsNCED4* mRNA, we have modified the description of “DRG9 can bind to stress-related mRNAs and lead them into SGs for protection against degradation” to “DRG9 can bind to *OsNCED4* mRNAs and lead them into SGs for protection against degradation” in line 426. In addition, we have also made modifications to similar descriptions in the title (line 3), abstract (line 31), introduction (line 90), text (line 188) and discussion (line 394).

6. Please check the Fig.1a and Extended Data Fig.1a for inconsistent horizontal coordinates. In Fig.1a, DRG9 was located in the interval of 15.241-15.244 Mb, while

in Extended Data Fig.1a, all 24 candidate genes were located in the interval of 15-15.2 Mb.

Response: We apologize for our negligence. We have now corrected this error (Extended Data Fig.1a).

7. Fig. 6a-f is not mentioned in the main text, and it should be referenced in line 303, line 307, line 312, and line 317, respectively.

Response: We apologize for the incorrect labeling, and have revised in line 320, line 324, line 329 and line 334.

8. In Fig.7b, it is suggested to avoid using red and green simultaneously for the pie charts.

Response: Thank you for your suggestion. We have made color modifications to the Fig.7b.

9. In line 887-888, the growth stage and tissues used for analyzing the expression levels of DRG9 should be described.

Response: Thank you for your suggestion. As shown in Extended Data Fig.1d,e legend, we described the growth stage and tissue of samples used for analyzing *DRG9* expression profiles.

10. The abstract could be modified to be more concise.

Response: Thank you for your suggestion. We have revised the abstract to make its content more concise.

11. In the part of discussion, I suggest that the value or advantage of the DRG9 gene in the application of rice breeding should be highlight.

Response: Thank you for your suggestion. We further discussed the advantages of *DRG9-DR* allele in rice breeding. *DRG9-DR* allele is generally lacking in the indica rice population. With the worsening global environment and frequent natural disasters such as drought in southern indica rice regions, utilizing the *DRG9-DR* allele to improve drought resistance in indica rice is of great significance for ensuring food security. We have revised the discussion in the new version (line 464-468).

Reviewers' Comments:

Reviewer #1:

Remarks to the Author:

In their revised manuscripts authors have considered some of my comments. There is an important and pending issue. In their rebuttal, authors considered *N. benthamiana* "membranes". I meant the membranes in *A. thaliana* root cells. I have attached the manuscript file denoting with arrows the places where I see localization-wetting on membranes.

Most importantly, this localization is reminiscent of what has been reported about the tug-of-war between membrane surfaces and condensates in Liu et al., 2023 (EMBO J). Furthermore, their treatment with CHX+MAN shows even more membranes, which supports the competition model. I assume, from their images, that their condensate has the propensity to wet membranes; in one of the images, they may even see a cell plate. In any case, this should be briefly discussed. Is there a "membrane binding" site (or equivalent, hydrophobic region) on DRG9?

Reviewer #2:

Remarks to the Author:

The authors have addressed my concerns and I do not have further comment.

Reviewer #3:

Remarks to the Author:

In this revised version of the manuscript, the authors have addressed most of the concerns I raised in my initial review. The new data and corrections have improved the manuscript. I think the following suggestions can be considered before it is accepted for publication.

1. The authors addressed that among the ORF4, ORF6, ORF16, ORF19, ORF22, and ORF23 knockout plants, only ORF23 had a drought-sensitive phenotype, but I couldn't find the contents in the current manuscript. I suggest adding that.
2. There are formatting errors that need to be corrected, such as, using hyphens for "protein-RNA" and "RT-qPCR" in line 592 and line 682, respectively, and using subscripts for "log₂FC" in line 622, etc.
3. In the section of references, the page number is incorrectly marked for most of the references. For instance, the page number of Reference2 is "16402-16409", not "16402-9". Please check that carefully.

Reviewer #1 (Remarks to the Author):

In their revised manuscripts authors have considered some of my comments. There is an important and pending issue. In their rebuttal, authors considered *N. benthamiana* "membranes". I meant the membranes in *A. thaliana* root cells. I have attached the manuscript file denoting with arrows the places where I see localization-wetting on membranes.

Most importantly, this localization is reminiscent of what has been reported about the tug-of-war between membrane surfaces and condensates in Liu et al., 2023 (EMBO J). Furthermore, their treatment with CHX+MAN shows even more membranes, which supports the competition model. I assume, from their images, that their condensate has the propensity to wet membranes; in one of the images, they may even see a cell plate. In any case, this should be briefly discussed. Is there a "membrane binding" site (or equivalent, hydrophobic region) on DRG9?

Based on your suggestion, we have predicted the membrane binding region and hydrophobic region of DRG9. As shown in following figure, the MBPPred prediction server (<http://aias.biol.uoa.gr/MBPpred/>) did not find significant membrane binding regions. However, the hydrophobic region prediction by ExPASy (<https://web.expasy.org/protscale/>) did find two regions (13-21 aa, 33-40 aa) with high hydrophobicity in the IDR (Score = 1.978 and 1.967 respectively), indicating that IDR may possess membrane association probability. We have cited the Liu et al, EMBO J 2023 paper and discussed this in the Discussion section (line 422-427).

MDFADAPSHGDSLVVVRDALLSQLQQDRLRKDIIVAE~~LA~~KIERAMALRDVVSQSPTR
HAAAAAAGKTITTVATPAKKPSPSEKSEPAVQKSMPPSAWSCAVCQVRTTSE~~RL~~RD
HCGGQKHQSKVAALEKTTKAMARTTAKPSPGAAARWGC~~SICNISCNGECDFD~~THLK
GKKHQANTQALLEQNKKSSVNPESQGTAAAATLICRV~~CQAKFTCQSDLQSHL~~KVM
KHQLNLRAPSSDGSSFTSATSESLSELYSCKVCSVKCTFERMLAYH~~LTGKKHL~~KQEN
LQLSCEICKLQCNSEKVLSDHRYGKKHQAKLEKVLQAKLNATE*~~↵~~

Reviewer #2 (Remarks to the Author):

The authors have addressed my concerns and I do not have further comment.

We thank you for your inputs which have helped us to improve the m/s.

Reviewer #3 (Remarks to the Author):

In this revised version of the manuscript, the authors have addressed most of the concerns I raised in my initial review. The new data and corrections have improved the manuscript. I think the following suggestions can be considered before it is accepted for publication.

1. The authors addressed that among the ORF4, ORF6, ORF16, ORF19, ORF22, and ORF23 knockout plants, only ORF23 had a drought-sensitive phenotype, but I couldn't find the contents in the current manuscript. I suggest adding that.

Based on your suggestion, we have added the genotype and phenotype data of these mutants to the new manuscript (Extended Data Fig. 1c,d). We have revised these results in the new manuscript (line108-113).

2. There are formatting errors that need to be corrected, such as, using hyphens for “protein-RNA” and “RT-qPCR” in line 592 and line 682, respectively, and using subscripts for “log₂FC” in line 622, etc.

Thanks for your suggestion, we have made the correction accordingly.

3. In the section of references, the page number is incorrectly marked for most of the references. For instance, the page number of Reference2 is “16402–16409”, not “16402-9”. Please check that carefully.

Thanks for your suggestion, we have made the correction accordingly.

Reviewers' Comments:

Reviewer #1:

Remarks to the Author:

The authors have addressed the remaining issue. I congratulate them for their work.

Reviewer #3:

Remarks to the Author:

The authors have addressed all my comments and suggestions. The manuscript has been greatly improved, and I believe that the great work will be important for future study. Congratulations for the authors.

Response to the comments of three reviewers

Reviewer #1 (Remarks to the Author):

The authors have addressed the remaining issue. I congratulate them for their work.

Response: Thanks for your comment.

Reviewer #3 (Remarks to the Author):

The authors have addressed all my comments and suggestions. The manuscript has been greatly improved, and I believe that the great work will be important for future study. Congratulations for the authors.

Response: Thanks for your comment.